# The Neurodata Without Borders ecosystem for neurophysiological data science

**Oliver Rübel**[1]*†, **Andrew Tritt**[2]†, **Ryan Ly**[1]†, **Benjamin K Dichter**[3]†, **Satrajit Ghosh**[4,5]†, **Lawrence Niu**[6], **Pamela Baker**[7], **Ivan Soltesz**[8], **Lydia Ng**[7], **Karel Svoboda**[7,9], **Loren Frank**[9,10,11], **Kristofer E Bouchard**[1,10,12,13,14]*

[1]Scientific Data Division, Lawrence Berkeley National Laboratory, Berkeley, United States; [2]Applied Mathematics and Computational Research Division, Lawrence Berkeley National Laboratory, Berkeley, United States; [3]CatalystNeuro, Benicia, United States; [4]McGovern Institute for Brain Research, Massachusetts Institute of Technology, Cambridge, United States; [5]Department of Otolaryngology - Head and Neck Surgery, Harvard Medical School, Boston, United States; [6]MBF Bioscience, Ashburn, United States; [7]Allen Institute for Brain Science, Seattle, United States; [8]Department of Neurosurgery, Stanford University, Stanford, United States; [9]Janelia Research Campus, Howard Hughes Medical Institute, Ashburn, United States; [10]Kavli Institute for Fundamental Neuroscience, San Francisco, United States; [11]Departments of Physiology and Psychiatry University of California, San Francisco, San Francisco, United States; [12]Biological Systems and Engineering Division, Lawrence Berkeley National Laboratory, Berkeley, United States; [13]Helen Wills Neuroscience Institute and Redwood Center for Theoretical Neuroscience, University of California, Berkeley, Berkeley, United States; [14]Weill Neurohub, Berkeley, United States

**\*For correspondence:**
oruebel@lbl.gov (OR);
kebouchard@lbl.gov (KEB)

†These authors contributed equally to this work

**Abstract** The neurophysiology of cells and tissues are monitored electrophysiologically and optically in diverse experiments and species, ranging from flies to humans. Understanding the brain requires integration of data across this diversity, and thus these data must be findable, accessible, interoperable, and reusable (FAIR). This requires a standard language for data and metadata that can coevolve with neuroscience. We describe design and implementation principles for a language for neurophysiology data. Our open-source software (Neurodata Without Borders, NWB) defines and modularizes the interdependent, yet separable, components of a data language. We demonstrate NWB's impact through unified description of neurophysiology data across diverse modalities and species. NWB exists in an ecosystem, which includes data management, analysis, visualization, and archive tools. Thus, the NWB data language enables reproduction, interchange, and reuse of diverse neurophysiology data. More broadly, the design principles of NWB are generally applicable to enhance discovery across biology through data FAIRness.

## Editor's evaluation

This manuscript provides an overview of an important project that proposes a common language to share neurophysiology data across diverse species and recording methods, Neurodata Without Borders (NWB). The NWB project includes tools for data management, analysis, visualization, and archiving, which are applicable throughout the context of the entire data lifecycle. This paper will help raise awareness of this endeavor and should be useful for many researchers across a broad range of fields who are interested in analyzing diverse neurophysiology datasets.

**eLife digest** The brain is an immensely complex organ which regulates many of the behaviors that animals need to survive. To understand how the brain works, scientists monitor and record brain activity under different conditions using a variety of experimental techniques. These neurophysiological studies are often conducted on multiple types of cells in the brain as well as a variety of species, ranging from mice to flies, or even frogs and worms.

Such a range of approaches provides us with highly informative, complementary 'views' of the brain. However, to form a complete, coherent picture of how the brain works, scientists need to be able to integrate all the data from these different experiments. For this to happen effectively, neurophysiology data need to meet certain criteria: namely, they must be findable, accessible, interoperable, and re-usable (or FAIR for short). However, the sheer diversity of neurophysiology experiments impedes the 'FAIR'-ness of the information obtained from them.

To overcome this problem, researchers need a standardized way to communicate their experiments and share their results – in other words, a 'standard language' to describe neurophysiology data. Rübel, Tritt, Ly, Dichter, Ghosh et al. therefore set out to create such a language that was not only FAIR, but could also co-evolve with neurophysiology research.

First, they produced a computer software program (called Neurodata Without Borders, or NWB for short) which generated and defined the different components of the new standard language. Then, other tools for data management were created to expand the NWB platform using the standardized language. This included data analysis and visualization methods, as well as an 'archive' to store and access data. Testing the new language and associated tools showed that they indeed allowed researchers to access, analyze, and share information from many different types of experiments, in organisms ranging from flies to humans.

The NWB software is open-source, meaning that anyone can obtain a copy and make changes to it. Thus, NWB and its associated resources provide the basis for a collaborative, community-based system for sharing neurophysiology data. Rübel et al. hope that NWB will inspire similar developments across other fields of biology that share similar levels of complexity with neurophysiology.

## Introduction

The immense diversity of life on Earth (*Darwin, 1909*) has always provided both inspiration and insight for biologists. For example, in neuroscience, the functioning of the brain is studied in species ranging from flies, to mice, to humans (*Figure 1a*; *Kandel et al., 2013*). Because brains evolved to produce a plethora of behaviors that advance organismal survival, neuroscientists monitor brain activity with a variety of different tasks and neural recording techniques. (*Figure 1a*). These technologies provide complementary views of the brain, and creating a coherent model of how the brain works will require synthesizing data generated by these heterogeneous experiments. However, the extreme heterogeneity of neurophysiological experiments impedes the integration, reproduction, interchange, and reuse of diverse neurophysiology data. As other fields of science, such as climate science (*Eaton, 2003*), astrophysics (*Hanisch et al., 2001*), and high-energy physics (*Brun and Rademakers, 1997*) have demonstrated, community-driven standards for data and metadata are a critical step in creating robust data and analysis ecosystems, as well as enabling collaboration and reuse of data across laboratories. A standardized language for neurophysiology data and metadata (i.e., a data language) is required to enable neuroscientists to effectively describe and communicate about their experiments, and thus share the data.

The extreme heterogeneity of neurophysiology experiments is exemplified in *Figure 1*. Diverse experiments are designed to investigate a variety of neural functions, including sensation, perception, cognition, and action. Tasks include running on balls or treadmills (e.g. pictures, *Figure 1i*; *Mallory et al., 2021*), memory-guided navigation of mazes (*Figure 1ii*; *Chung et al., 2019*), production of speech (*Figure 1iii*; *Bouchard et al., 2013*), and memory formation (*Figure 1iv*). The use of different species in neuroscience is driven, in part, by the applicability of specific neurophysiological recording techniques (*Figure 1b*). For example, the availability of genetically modified mice makes this species ideal to monitor the activity of genetically defined neurons using calcium sensors (e.g. with GCaMP; optophysiology, 'o-phys'; *Figure 1bi, ci*). On the other hand, intracranially implanted

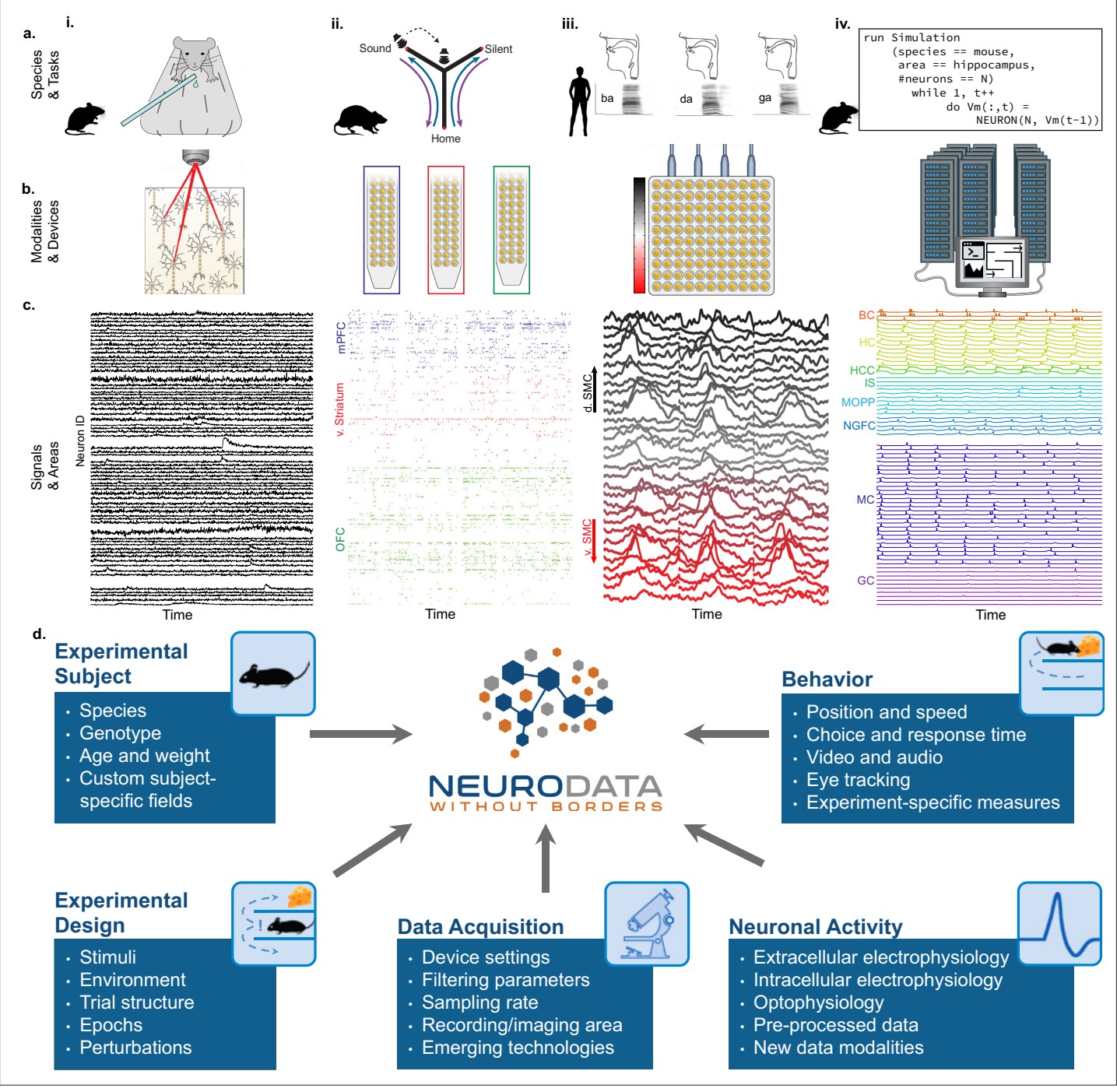

**Figure 1.** NWB addresses the massive diversity of neurophysiology data and metadata. (**a**) Diversity of experimental systems: species and tasks: (i) mice performing a visual discrimination task; (ii) rats performing a memory-guided navigation task; (iii) humans speaking consonant-vowel syllables; (iv) biophysically detailed simulations of mouse hippocampus during memory formation. The corresponding acquisition modalities and signals are shown in the corresponding columns in figure (**b and c**). (**b**) Diversity of data modalities and acquisition devices: (i) optophysiological Ca²⁺ imaging with two-photon microscope; (ii) intra-cortical extracellular electrophysiological recordings with polytrodes in multiple brain areas (indicated by color, see c.ii); (iii) cortical surface electrophysiology recordings with electrocorticography grids; (iv) high-performance computing systems for large-scale, biophysically detailed simulations of large neural networks. (**c**) Diversity of signals and areas: (i) Ca²⁺ signals as a function of time from visually identified individual neurons in primary visual cortex (V1) (*Mallory et al., 2021*); (ii) spike-raster (each tick demarcates the time of an action potential) from simultaneously recorded putative single-units after spike-sorting of extracellular signals from medial prefrontal cortex (mPFC; blue), ventral striatum (v. Striatum, red), and orbital frontal cortex (OFC, green) (color corresponds to b.ii) (*Chung et al., 2019*); (iii) high-gamma band activity from electrodes over the speech sensorimotor cortex (SMC), with dorsal-ventral distance from Sylvian fissure color coded red-to-black (color corresponds to

*Figure 1 continued*

b.iii) (***Bouchard et al., 2013***); (iv) simulated intracellular membrane potentials from different cell-types from large-scale biophysical simulation of the hippocampus (BC, Basket Cell); HC, Hilar Interneuron (with axon associated with the) Perforant Path; HCC, Hilar Interneuron (with axon associated with the) Commissural/Associational Path; IS, Interneuron-Specific Interneuron; MCPP, medial Perforant Path; NGFC, neurogliaform cell; MC, mossy cell; GC, granule cell](Raikov and Soltesz, unpublished data). (**d**) Neurodata Without Borders (NWB) provides a robust, extensible, and maintainable software ecosystem for standardized description, storage, and sharing of the diversity of experimental subjects, behaviors, experimental designs, data acquisition systems, and measures of neural activity exemplified in **a – c**.

electrophysiology probes ('e-phys') with large numbers of electrodes enable monitoring the activity of many single neurons at millisecond resolution from different brain regions simultaneously in freely behaving rats (***Figure 1bii, cii***; ***Chung et al., 2019***). Likewise, in human epilepsy patients, arrays of electrodes on the cortical surface (i.e. electrocorticography, ECoG) provides direct electrical recording of mesoscale neural activity at high-temporal resolution across multiple brain areas (e.g., speech sensorimotor cortex 'SMC'; ***Figure 1biii, ciii***; ***Bouchard et al., 2013***). Additionally, to understand the intracellular functioning of single neurons, scientists measure membrane potentials (ic-ephys), for example, via patch clamp recordings (see Appendix 1). As a final example, to study the detailed workings of complete neural circuits, supercomputers are used for biophysically detailed simulation of the intracellular membrane potentials of a large variety of neurons organized in complex networks (***Bezaire et al., 2016***; Raikov and Soltesz, unpublished data; ***Figure 1biv, civ***).

Although the heterogeneity described above is most evident across labs, it is present in a reduced form within single labs; lab members can use new equipment or different techniques in custom experiments to address specific hypotheses. As such, even within the same laboratory, storage and descriptions of data and metadata often vary greatly between experiments, making archival sharing and reuse of data a significant challenge. Across species and tasks, different acquisition technologies measure different neurophysiological quantities from multiple spatial locations over time. Thus, the numerical data itself can commonly be described in the form of space-by-time matrices, the storage of which has been optimized (for space and rapid access) by computer scientists for decades. It is the immense diversity of metadata required to turn those numbers into knowledge that presents the outstanding challenge.

Scientific data must be thought of in the context of the entire data lifecycle, which spans planning, acquisition, processing, and analysis to publication and reuse (***Griffin et al., 2017***). In this context, a 'data ecosystem' is a shared market for scientific data, software, and services that are able to work together. Such an ecosystem for neurophysiology would empower users to integrate software components and products from across the ecosystem to address complex scientific challenges. Foundational to realizing a data ecosystem is a common 'language' that enables seamless exchange of data and information between software components and users. Here, the principles of Findable, Accessible, Interoperable, and Reusable (i.e. FAIR) (***Wilkinson et al., 2016***) data management and stewardship are widely accepted as essential to ensure that data can flow reliably between the components of a data ecosystem. Traditionally, data standards are often understood as rigid and static data models and formats. Such standards are particularly useful to enable the exchange of specific data types (e.g. image data), but are insufficient to address the diversity of data types generated by constantly evolving experiments. Together, these challenges and requirements necessitate a conceptual departure from the traditional notion of a rigid and static data standard. That is, we need a 'language' where fundamental structures can be reused and combined in new ways to express novel concepts and experiments. A data language for neurophysiology will enable precise communication about neural data that can co-evolve with the needs of the neuroscience community.

We created the Neurodata Without Borders (NWB) data language (i.e. a standardized language for describing data) for neurophysiology to address the challenges described above. NWB(v2) accommodates the massive heterogeneity and evolution of neurophysiology data and metadata in a unified framework through the development of a novel data language that can co-evolve with neurophysiology experiments. We demonstrate this through the storage of multimodal neurophysiology data, and derived products, in a single NWB file with easy visualization tools. This generality was enabled by the development of a robust, extensible, and sustainable software architecture based on our Hierarchical Data Modeling Framework (HDMF) (***Tritt et al., 2019***). To facilitate new experimental paradigms, we developed methods for creating and sharing NWB Extensions that permit the NWB data

language to co-evolve with the needs of the community. NWB is foundational for the Distributed Archives for Neurophysiology Data Integration (DANDI) data repository to enable collaborative data sharing and analysis. Together, NWB and DANDI make neurophysiology data FAIR. Indeed, NWB is integrated with a growing ecosystem of state-of-the-art analysis tools to provide a unified storage platform throughout the data life cycle. Through extensive and coordinated efforts in community engagement, software development, and interdisciplinary governance, NWB is now being utilized by more than 53 labs and research organizations. Across these groups, NWB is used for all neurophysiology data modalities collected from species ranging from flies to humans during diverse tasks. Together, the capabilities of NWB provide the basis for a community-based neurophysiology data ecosystem. The processes and principles we utilized to create NWB provide an exemplar for biological data ecosystems more broadly.

## Results

### NWB enables unified description and storage of multimodal data and derived products

NWB files contain all of the measurements for a single experiment, along with all of the necessary metadata to understand that data. Neurophysiology experiments often contain multiple simultaneous streams of data, for example, via simultaneous recording of neural activity, sensory stimuli, behavioral tracking, and direct neural modulation. Furthermore, neuroscientists are increasingly leveraging multiple neurophysiology recording modalities simultaneously (e.g. ephys and ophys), which offer complementary information not achievable in a single modality. These distinct raw data input types often require processing, further expanding the multiplicity of data types that need to be described and stored.

A key capability of NWB is to describe and store many data sources (including neurophysiological recordings, behavior, and stimulation information) in a unified way that is readily analyzed with all time bases aligned to a common clock. For each data source, raw acquired signals and/or preprocessed data can be stored in the same file. *Figure 2a* illustrates a workflow for storing and processing electrophysiology and optical physiology in NWB (*Ledochowitsch et al., 2019*; *Huang et al., 2020*). Raw voltage traces (*Figure 2a*, top) from an extracellular electrophysiology recording and image sequences from an optical recording (*Figure 2a*, bottom) can both be stored in the same NWB file, or separate NWB files synchronized to each other. Extracellular electrophysiology data often goes through spike sorting, which processes the voltage traces into putative single units and action potential (a.k.a., spikes) times for those units (*Figure 2a*, top). The single unit spike times can then also be written to the NWB file. Similarly, optical physiology is generally processed using segmentation algorithms to identify regions of the image that correspond to neurons and extract fluorescence traces for each neuron (*Figure 2a*, bottom). The fluorescence traces can also be stored in the NWB file, resulting in raw and processed data for multiple input streams. The timing of these streams is each defined separately, allowing streams with different sampling rates and starting times to be registered to the same common clock. As illustrated in *Figure 1d*, NWB can also store raw and processed behavioral data as well as stimuli, such as animal location and amplitude/frequency of sounds. The multi-modal capability of NWB is critical for capturing the diverse types of data simultaneously acquired in many neurophysiology experiments, particularly if those experiments involve multiple simultaneous neural recording modalities.

Having pre-synchronized data in the same format enables faster and less error-prone development of analysis and visualizations tools that provide simultaneous views across multiple streams. *Figure 2b* shows an interactive dashboard for exploring a dataset of simultaneously recorded optical physiology and electrophysiology data published by the Allen Institute (*Ledochowitsch et al., 2019*). This dashboard illustrates the simultaneous exploration of five data elements all stored in a single NWB file. The microscopic image panel (*Figure 2b*, far left) shows a frame of the video recorded by the microscope. The red outline overlaid on that image shows the region-of-interest where a cell has been identified by the experimenter. The fluorescence trace (dF/F) shows the activation of the region-of-interest over time. This activity is displayed in line with electrophysiology recordings of the same cell (ephys), and extracted spikes (below ephys). Interactive controls (*Figure 2b*, bottom) allow a user to explore the complex and important relationship between these data sources.

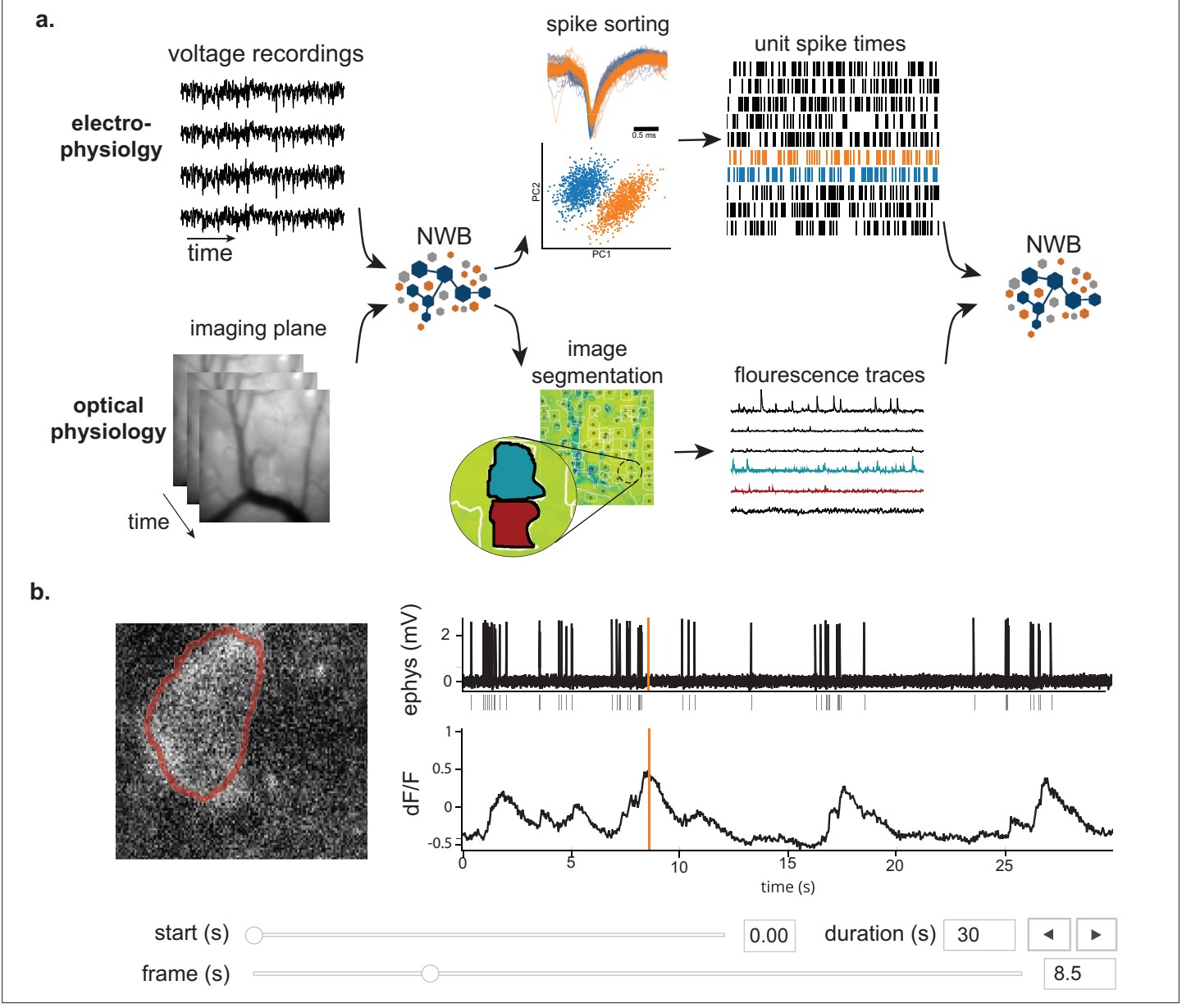

**Figure 2.** NWB enables unified description and storage of multimodal raw and processed data. (**a**) Example pipelines for extracellular electrophysiology and optical physiology demonstrate how NWB facilitates data processing. For extracellular electrophysiology (top), raw acquired data is written to the NWB file. The NWB ecosystem provides interfaces to a variety of spike sorters that extract unit spike times from the raw electrophysiology data. The spike sorting results are then stored into the same NWB file (bottom). Separate experimental data acquired from an optical technique is converted and written to the NWB file. Several modern software tools can then be used to process and segment this data, identifying regions that correspond to individual neurons, and outputting the fluorescence trace of each putative neuron. The fluorescence traces are written to the same NWB file. NWB handles the time alignment of multiple modalities, and can store multiple modalities simultaneously, as shown here. The NWB file also contains essential metadata about the experimental preparation. (**b**) NWBWidgets provides visualizations for the data within NWB files with interactive views of the data across temporally aligned data types. Here, we show an example dashboard for simultaneously recorded electrophysiology and imaging data. This interactive dashboard shows on the left the acquired image and the outline of a segmented neuron (red) and on the right a juxtaposition of extracellular electrophysiology, extracted spike times, and simultaneous fluorescence for the segmented region. The orange line on the ephys and dF/F plots indicate the frame that is shown to the left. The controls shown at the bottom allow a user to change the window of view and the frame of reference within that window.

Visualizations of multiple streams of data is a common need across different types of neurophysiology data. Another NWBWidgets dashboard is described in *Peterson et al., 2021*, which demonstrates a dashboard for viewing human body position tracking with simultaneously acquired ECoG data, as well as a panel for viewing the 3D position of electrodes on the participant's brain. Dashboards for specific experiment types can be constructed using NWBWidgets, a library for interactive web-based visualizations of NWB data, that provides tools for navigating and combining data across modalities.

## The NWB software architecture modularizes and integrates all components of a data language

Neuroscientists use NWB through a core software stack (*Figure 3b*) with four modularized components: the specification language, the data standard schema, data use APIs, and storage backends (*Figure 3a*) The identification and modularization of these components was a core conceptual advance of the NWB software. This software architecture provides flexible accommodation of the heterogenous use cases and needs of NWB users. Modularizing the software in this way allows extending the schema to handle new types of data, to implement APIs in new programming languages, and to store NWB using different backends, all while maintaining compliance with NWB and providing a stable interface for users to interact with.

First, we describe the specification language used to define hierarchical data models. The YAML-based specification language defines four primitive structures: Groups, Datasets, Attributes, and Links. Each of these primitive structures has characteristics to define their names and parameters (e.g. the allowable shapes of a Dataset). Importantly, these primitives are abstract, and are not tied to any particular data storage backend. The specification language also uses object-oriented principles to define neurodata types that, like classes, can be reused through inheritance and combined through composition to build more complex structures.

The NWB core schema uses the primitives defined in the specification language to define more complicated structures and requirements for particular types of neurophysiology data. For instance, an ElectricalSeries is a neurodata type that defines the data and metadata for an intracranially recorded voltage time series in an extracellular electrophysiology experiment. ElectricalSeries extends the TimeSeries neurodata type, which is a generic structure designed for any measurement that is sampled over time, and defines fields, such as, data, units of measurement, and sample times (specified either with timestamps or sampling rate and start time). ElectricalSeries also requires an electrodes field, which provides a reference to a table of electrodes describing the locations and characteristics of the electrodes used to record the data. The NWB core schema defines many neurodata types in this way, building from generic concepts to specific data elements. The neurodata types have rigorous metadata requirements that ensure a sufficiently rich description of the data for reanalysis. The neurodata types are divided into modules such as ecephys (extracellular electrophysiology), icephys (intracellular electrophysiology), ophys (optical physiology), and behavior. Importantly, the core schema is defined on its own and is agnostic to APIs and programming languages. This allows for the creation of an API in any programming language, which will allow NWB to stay up to date as programming technologies advance.

Application Programming Interfaces (APIs) allow convenient interfaces for writing and reading data according to the NWB schema. The development team maintains APIs in Python (PyNWB) and MATLAB (MatNWB), the two most widely used programming languages in neurophysiology. These APIs (*Figure 3c and d*) are governed by the NWB schema and use an object-oriented design in which neurodata types (e.g. ElectricalSeries or TowPhotonSeries) are represented by a dedicated interface class. Both APIs are fully compliant with the NWB standard and are, hence, interoperable (i.e. files generated by PyNWB can be read via MatNWB and vice versa). Both APIs also support advanced data Input/Output (I/O) features, such as lazy data read, compression, and iterative data write for data streaming. A key difference in the design of PyNWB and MatNWB is the implementation of the data translation process. PyNWB uses a dynamic data translation process based on data builders (*Figure 3c*). The data builders are classes that mirror the NWB specification language primitives and provide an interoperability layer where data from different storage backends can be mapped using object mappers into a uniform API. In contrast, MatNWB implements a static translation process that generates the MATLAB API classes automatically from the schema (*Figure 3d*). The MatNWB

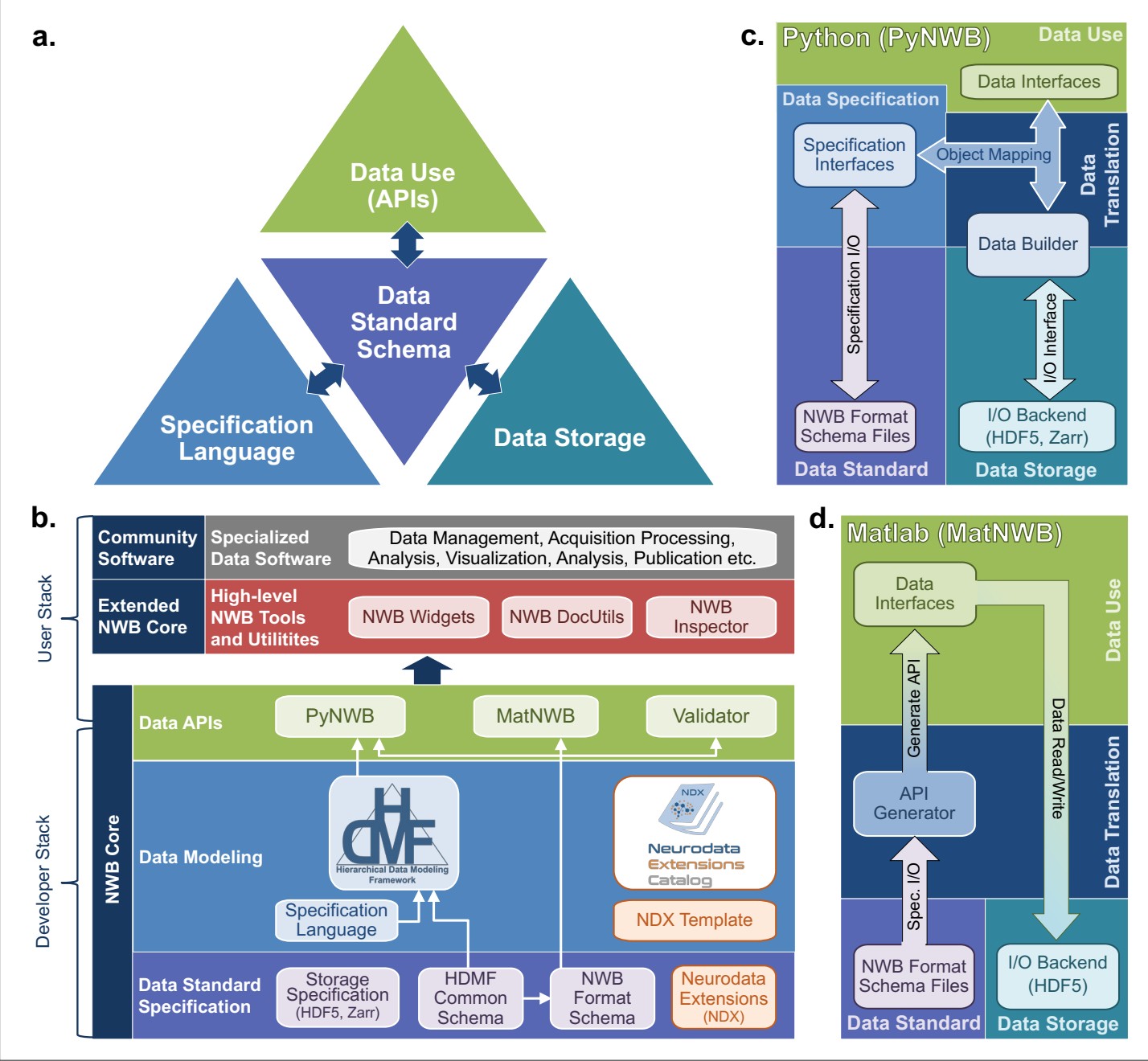

**Figure 3.** The NWB software architecture modularizes and integrates all components of a data language. (**a**) Illustration of the main components of the NWB software stack consisting of: (i) the specification language (light blue) to describe data standards, (ii) the data standard schema (lilac), which uses the specification language to formally define the data standard, (iii) the data storage (blue gray) for translating the data primitives (e.g., groups and datasets) described by the schema to/from disk, and (iv) the APIs (green) to enable users to easily read and write data using the standard. Additional data translation components (dark blue arrows) defined in the software then insulate and separate these four main components to enable the individual components to evolve while minimizing impacts on the other components. For example, by insulating the schema from the storage we can extend the standard schema without having to modify the data storage and conversely also integrate new storage backends without having to modify the standard schema. (**b**) Software stack for defining and extending the NWB data standard and creating and using NWB data files. The software stack covers all aspects of data standardization: (i) data specification, (ii) data modeling, (iii) data storage, (iv) data APIs, (v) data translation, and (vi) tools. Depending on their role, different stakeholders typically interact with different subsets of the software ecosystem. End users typically interact with the data APIs (green) and higher-level tools (red, gray) while tool developers typically interact with the data APIs and data modeling layers (green, blue). Working groups and developers of extensions then typically interact with the data modeling and data standard specification components. Finally, core NWB developers typically interact with the entire developer stack, from foundational documents (lilac) to data APIs (green). (**c**) Software architecture of

*Figure 3 continued on next page*

*Figure 3 continued*

the PyNWB Python API. PyNWB provides interfaces for interacting with the specification language and schema, data builders, storage backends, and data interfaces. Additional software components (arrows) insulate and formalize the transitions between the various components. The object-mapping-based data translation describes: (i) the integration of data interfaces (which describe the data) with the specification (which describes the data model) to generate data builders (which describe the data for storage) and (ii) vice versa, the integration of data builders with the specification to create data interfaces. The object mapping insulates the end-users from specifics of the standard specification, builders, and storage, hence, providing stable, easy-to-use interfaces for data use that are agnostic of the data storage and schema. The I/O interface then provides an abstract interface for translating data builders to storage which specific I/O backends must implement. Finally, the specification I/O then describes the translation of schema files to/from storage, insulating the specification interfaces from schema storage details. Most of the data modeling, data translation, and data storage components are general and implemented in HDMF. This approach facilitates the application of the general data modeling capabilities we developed to other science applications and allows PyNWB itself to focus on the definition of data interfaces and functionality that are specific to NWB. (**d**) Software architecture of the MatNWB Matlab API. MatNWB generates front-end data interfaces for all NWB types directly from the NWB format schema. This allows MatNWB to easily support updates and extensions to the schema while enabling development of higher-level convenience functions.

approach simplifies updating of the API to support new versions of the NWB schema and extensions, and helps minimize cost for development, but with reduced flexibility in supported storage backend and API. The difference in the data translation process between the APIs (i.e. static vs. dynamic) is a reflection of the different target uses (*Figure 3c and d*). MatNWB primarily targets data conversion and analysis. In contrast, PyNWB additionally targets integration with data archives and web technologies and is used heavily for development of extensions and exploration of new technologies, such as alternate storage backends and parallel computing libraries.

Finally, the specification of data storage backends deals with translating NWB data models to/from storage on disk. Data storage is governed by formal specifications describing the translation of NWB data primitives (e.g. groups or datasets) to primitives of the particular storage backend format (e.g. HDF5) and is implemented as part of the NWB user APIs. HDF5 is our primary backend, chosen for its broad support across scientific programming languages, its sophisticated tools for handling large datasets, and its ability to express very complex hierarchical structures in relatively few files. The interoperability afforded by the PyNWB builders allows for other backends, and we have a prototype for storing NWB in the Zarr format.

Together, these four components (specification language, standard schema, APIs, and storage backend) and the interaction between them constitute a sophisticated software infrastructure that is applicable beyond neuroscience and could be useful to many other domains. Therefore, we have factored out the domain-agnostic components of each of these four components into a Python software package called the Hierarchical Data Modeling Framework (HDMF) (*Figure 3b*). Much of the infrastructure described here, including the specification language, fundamental structures of the core schema, base classes for the object mapper and builder layers, and base classes of the PyNWB API are defined in the HDMF package. With its modular architecture and open-source model, the NWB software stack instantiates the NWB data language and makes NWB accessible to users and developers. The NWB software design illustrates the complexity of creating a data language and provides reusable components (e.g. HDMF and the HDMF Common Schema) that can be applied more broadly to facilitate development of data languages for other biological fields in the future.

All NWB software is open source, managed and versioned using Git, and released using a permissive BSD license via GitHub. NWB uses automated continuous integration for testing on all major architectures (MacOS, Windows, and Linux) and all core software can be installed via common package managers (e.g. pip and conda). The suite of NWB core software tools (*Figure 3b*) enables users to easily read and write NWB files, extend NWB to integrate new data types, and builds the foundation for integration of NWB with community software. NWB data can also be easily accessed in other programming languages (e.g. IGOR or R) using the HDF5 APIs available across modern scientific programming languages.

## NWB enables creation and sharing of extensions to incorporate new use cases

As with all of biology, neurophysiological discovery is driven in large part by new tools that can answer previously unconsidered questions. Thus, a language for neurophysiology data must be able to co-evolve with the experiments being performed and provide customization capability while

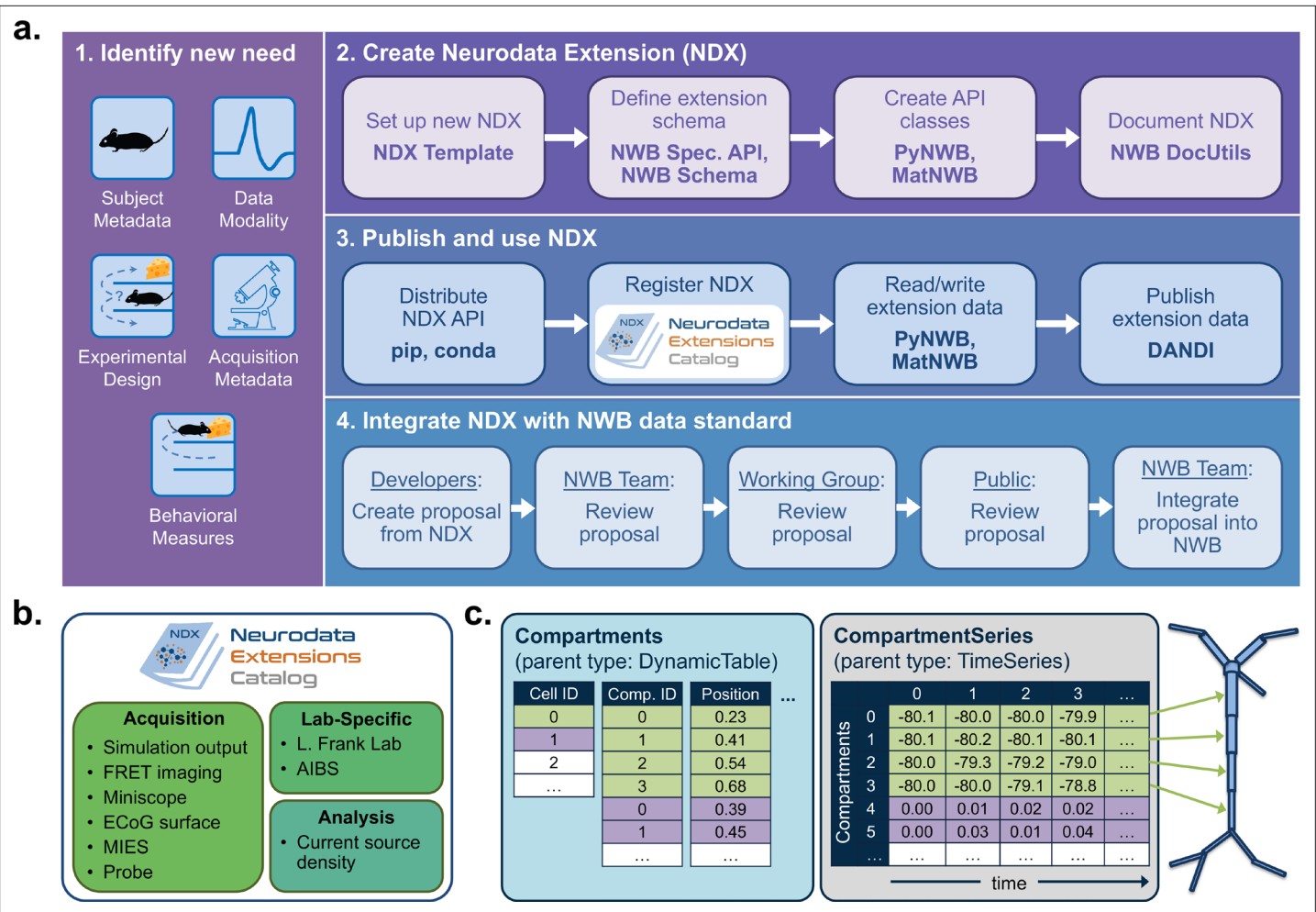

**Figure 4.** NWB enables creation and sharing of extensions to incorporate new use cases. (**a**) Schematic of the process of creating a new neurodata extension (NDX), sharing it, and integrating it with the core NWB data standard. Users first identify the need for a new data type, such as additional subject metadata or data from a new data modality. Users can then use the NDX Template, NWB Specification API, PyNWB/MatNWB data APIs, and NWB DocUtils tools to set up a new NDX, define the extension schema, define and test custom API classes for interacting with extension data, and generate Sphinx-based documentation in common formats, for example, HTML or PDF. After the NDX is completed, users can publish the NDX on PyPI and conda-forge for distribution via the pip and conda tools, and share extensions via the NDX Catalog, a central, searchable catalog. Users can easily read/write extension data using PyNWB/MatNWB and publish extension data in DANDI and other archives. Finally, extensions are used to facilitate enhancement, maintenance, and governance of the NWB data standard. Users may propose the integration of an extension published in the NDX Catalog with the core standard. The proposal undergoes three phases of review: an initial review by the NWB technology team, an evaluation by a dedicated working group, and an open, public review by the broader community. Once approved, the proposal is integrated with NWB and included in an upcoming version release. (**b**) Sampling of extensions currently registered in the NDX catalog. Users can search extensions based on keywords and textual descriptions of extensions. The catalog manages basic metadata about extensions, enabling users to discover and access extensions, comment and make suggestions, contribute to the source code, and collaborate on a proposal for integration into the core standard. While some extensions have broad applicability, others represent data and metadata for a specific lab or experiment. (**c**) Example extension for storing simulation output data using the SONATA framework. The new Compartments type extends the base DynamicTable type and contains metadata about each cell and compartment within each cell, such as position and label. The CompartmentSeries type extends the base TimeSeries type and contains a link to the Compartments type to associate each row of its data array with a compartment from the Compartments table.

maintaining stability. NWB enables the creation and sharing of user-defined extensions to the standard that support new and specialized data types (*Figure 4a1*). Neurodata Extensions (NDX) are defined using the same formal specification language used by the core NWB schema. Extensions can build off of data types defined in the core schema or other extensions through inheritance and composition. This enables the reuse of definitions and associated code, facilitates the integration with existing tools, and makes it easier to contextualize new data types.

NWB provides a comprehensive set of tools and services for developing and using neurodata extensions. The NWB Specification API, HDMF DocUtils, and the PyNWB and MatNWB user APIs work with extensions with little adjustment (*Figure 4a2*). In addition, the NDX Template makes it easy for users to develop new extensions. Appendix 2 demonstrates the steps outlined in *Figure 4* for the *ndx-simulation-output* extension shown in *Figure 4c*. The Neurodata Extensions Catalog (*Figure 4a3*) then provides a centralized listing of extensions for users to publish, share, find, and discuss extensions across the community. Appendix 3 provides a more detailed overview of the extension workflow as part of the NDX Catalog. Several extensions have been registered in the Neurodata Extensions Catalog (*Figure 4*), including extensions to support the storage of the cortical surface mesh of an electrocorticography experiment subject, storage of fluorescence resonance energy transfer (FRET) microscopy data, and metadata from an intracellular electrophysiology acquisition system. The catalog also includes the ndx-simulation-output extension for the storage of the outputs of large-scale simulations. Large-scale network models that are described using the new SONATA format (*Dai et al., 2020*) can be converted to NWB using this extension. The breadth of these extensions demonstrates that NWB will be able to accommodate new experimental paradigms in the future.

As particular extensions gain traction within the community, they may be integrated into the core NWB format for broader use and standardization (*Figure 4a4*). NWB has a formal, community-driven review process for refining the core format so that NWB can adapt to evolving data needs in neuroscience. The owners of the extension can submit a community proposal for the extension to the NWB Technical Advisory Board, which then evaluates the extension against a set of metrics and best practices published on the catalog website. The extension is then tested and reviewed by both a dedicated working group of potential stakeholders and the general public before it is approved and integrated into the core NWB format. Key advantages of the extension approach are to allow iterative development of extensions and complete implementation and vetting of new data types under several use cases before they become part of the core NWB format. The NWB extension mechanism thus enables NWB to provide a unified data language for all data related to an experiment, allows describing of data from novel experiments, and supports the process of evolving the core NWB standard to fit the needs of the neuroscientific community.

## NWB is foundational for the DANDI data repository to enable collaborative data sharing

Making neurophysiology data accessible supports published findings and allows secondary reuse. To date, many neurophysiology datasets have been deposited into a diverse set of repositories (e.g. CRCNS, Figshare, Open Science Framework, Gin). However, no single data archive provides the neuroscientific community the capacity and the domain specificity to store and access large neurophysiology datasets. Most current repositories have specific limits on data sizes and are often generic, and therefore lack the ability to search using domain specific metadata. Further, for most neuroscientists, these archives often serve as endpoints associated with publishing, while research is typically an ongoing and collaborative process. Few data archives support a collaborative research model that allows data submission prospectively, analysis of data directly in the archive, and opening the conversation to a broader community. Enabling reanalysis of published data was a key challenge identified by the BRAIN Initiative. Together, these issues impede access and reuse of data, ultimately decreasing the return on investment into data collection by both the experimentalist and the funding agencies.

To address these and other challenges associated with neurophysiology data storage and access, we developed DANDI, a Web-based data archive that also serves as a collaboration space for neurophysiology projects (*Figure 5*). The DANDI data archive (https://dandiarchive.org) is a cloud-based repository for cellular neurophysiology data and uses NWB as its core data language (*Figure 5b*). Users can organize collections of NWB files (e.g. recorded from multiple sessions) into DANDI datasets (so called Dandisets). Users can view the public Dandisets using a Web browser (*Figure 5a*) and search for data from different projects, people, species, and modalities. This search is over metadata that has been extracted directly from the NWB files where possible. Users can interact with the data in the archive using a JupyterHub Web interface (*Figure 5c*) to explore, visualize and analyze data stored in the archive. Using the DANDI Python client, users can organize data locally into the structure required by DANDI as well as download data from and upload data to the archive (*Figure 5d*). Software developers can access information about Dandisets and all the files it contains using the DANDI

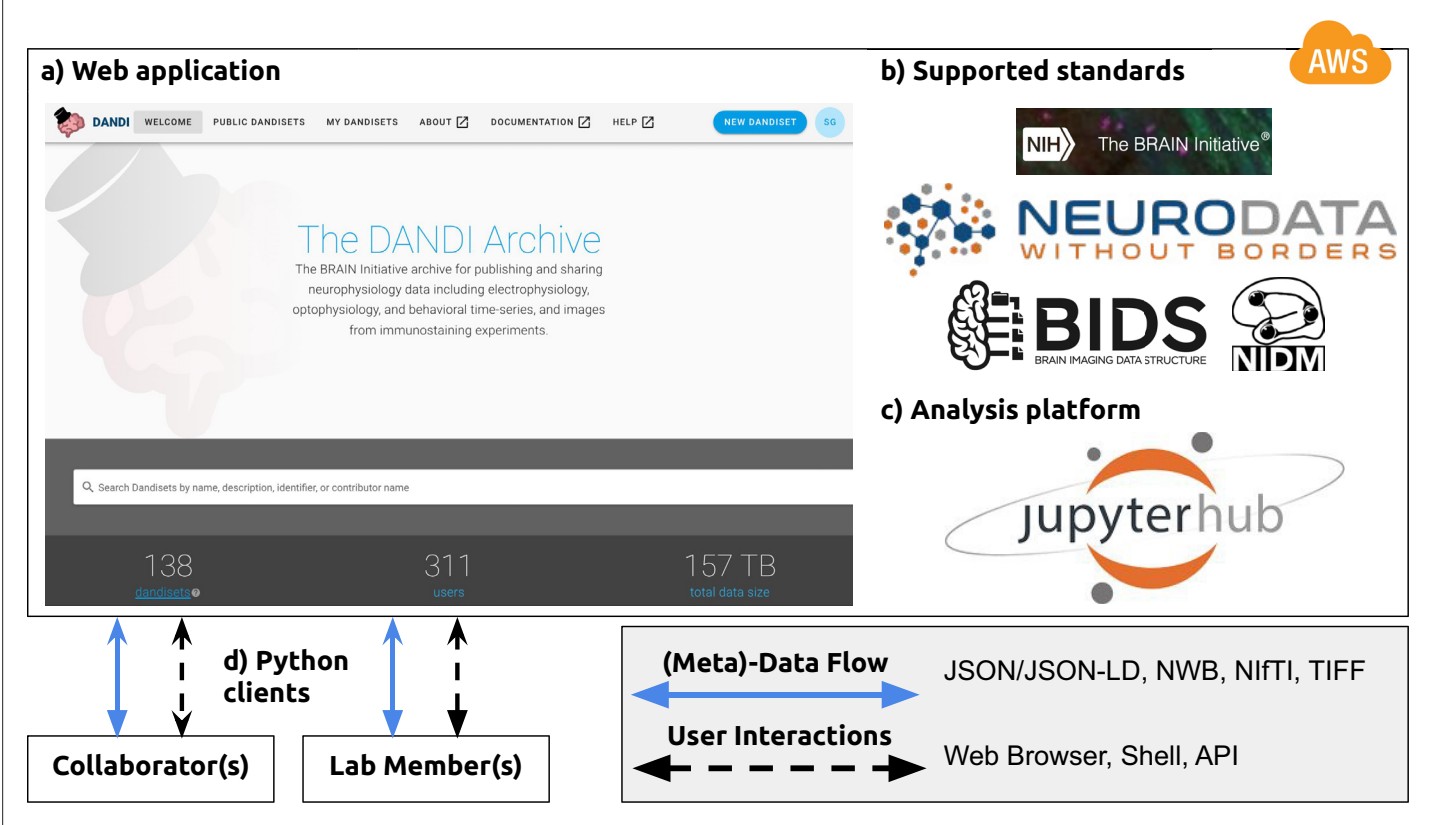

**Figure 5.** NWB is foundational for the DANDI data repository to enable collaborative data sharing. The DANDI project makes data and software for cellular neurophysiology FAIR. DANDI stores electrical and optical cellular neurophysiology recordings and associated MRI and/or optical imaging data. NWB is foundational for the DANDI data repository to enable collaborative data sharing. (**a**) DANDI provides a Web application allowing scientists to share, collaborate, and process data from cellular neurophysiology experiments. The dashboard provides a summary of Dandisets and allows users to view details of each dataset. (**b**) DANDI works with US BRAIN Initiative awardees and the neurophysiology community to curate data using community data standards such as NWB, BIDS, and NIDM. DANDI is supported by the US BRAIN Initiative and the Amazon Web Services (AWS) Public Dataset Program. (**c**) DANDI provides a JupyterHub interface to visualize the data and interact with the archive directly through a browser, without the need to download any data locally. (**d**) Using Python clients and/or a Web browser, researchers can submit and retrieve standardized data and metadata from the archive. The data and metadata use standard formats such as HDF5, JSON, JSON-LD, NWB, NIfTI, and TIFF.

Representational State Transfer (REST) API (https://api.dandiarchive.org/). The REST API also allows developers to create software tools and database systems that interact with the archive. Each Dandiest is structured by grouping files belonging to different biosamples, with some relevant metadata stored in the name of each file, and thus aligning itself with the BIDS standard (*Gorgolewski et al., 2016*). Metadata in DANDI is stored using the JSON-LD format, thus allowing graph-based queries and exposing DANDI to Google Dataset Search. Dandiset creators can use DANDI as a living repository and continue to add data and analyses to an existing Dandiset. Released versions of Dandisets are immutable and receive a digital object identifier (DOI). The data are presently stored on an Amazon Web Services Public dataset program bucket, enabling open access to the data over the Web, and backed up on institutional repositories. DANDI is working with hardware platforms (e.g. OpenEphys), database software (e.g. DataJoint) and various data producers to generate and distribute NWB datasets to the scientific community.

DANDI provides neuroscientists and software developers with a Platform as a Service (PAAS) infrastructure based on the NWB data language and supports interaction via the web browser or through programmatic clients, software, and other services. In addition to serving as a data archive and providing persistence to data, it supports continued evolution of Dandisets. This allows scientists to use the archive to collect data toward common projects across sites, and engage collaborators actively, directly at the onset of data collection rather than at the end. DANDI also provides a computational interface to the archive to accelerate analytics on data and links these Dandisets to

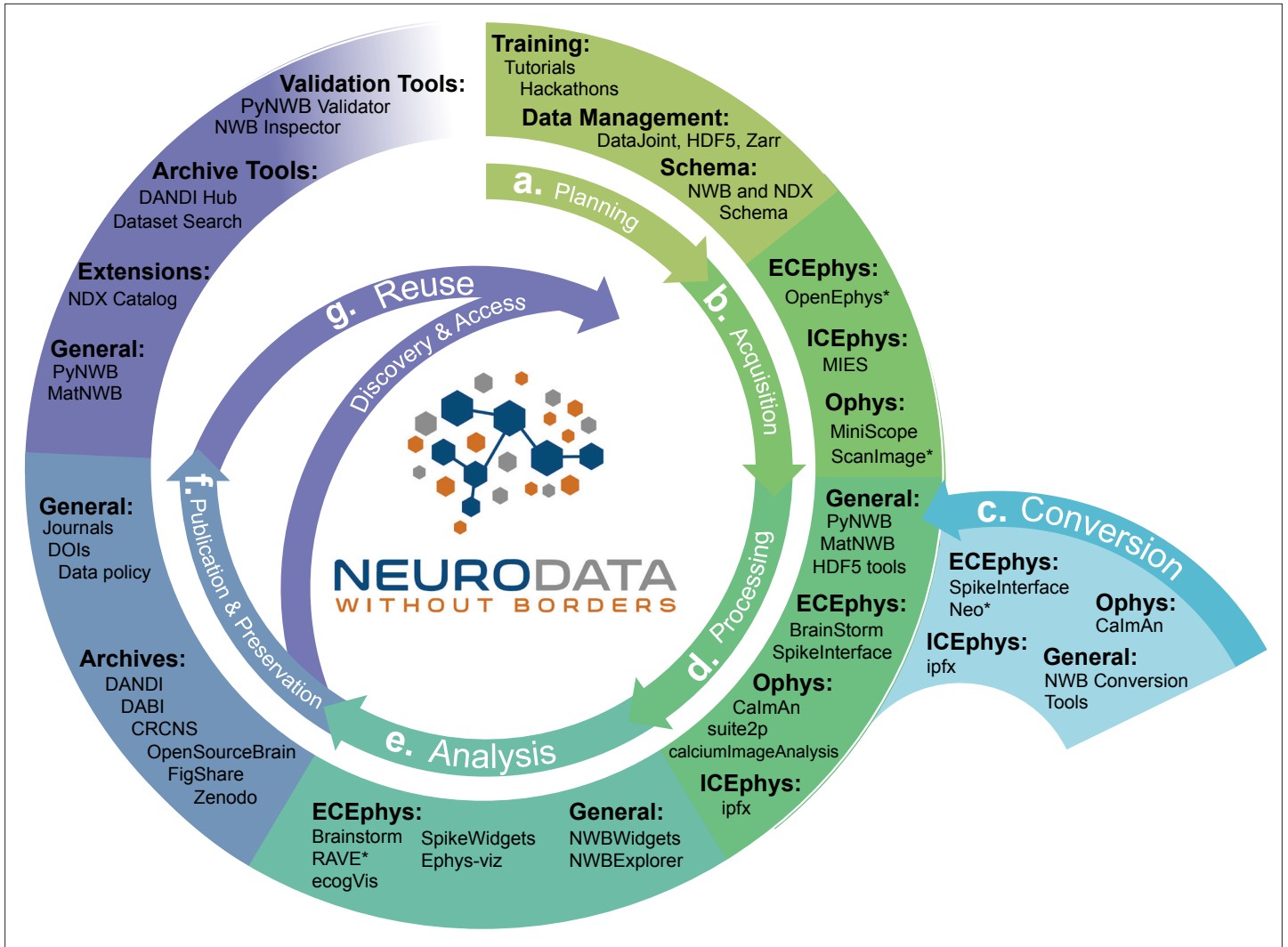

**Figure 6.** NWB is integrated with state-of-the-art analysis tools throughout the data life cycle. NWB technologies are at the heart of the neurodata lifecycle and applications. Data standards are a critical conduit that facilitate the flow of data throughout the data lifecycle and integration of data and software across all phases (a. to g.) of the data lifecycle. (**a**) NWB supports experimental planning through integration with data management, best practices, and by allowing users to clearly define what metadata to collect. (**b–c**) NWB supports storage of unprocessed acquired electrical and optical physiology signals, facilitating integration already during data acquisition. NWB is already supported by several acquisition systems (**b**) as well as a growing set of tools for conversion (**c**) of existing data to NWB. (**d**) Despite its young age, NWB is already supported by a large set of neurophysiology processing software and tools. Being able to access and evaluate multiple processing methods, e.g., different spike sorting algorithms and ROI segmentation methods, is important to enable high-quality data analysis. Through integration with multiple different tools, NWB provides access to broad range of spike sorters, including, MountainSort, KiloSort, WaveClust, and others, and ophys segmentation methods, e.g., CELLMax, CNMF, CNMF-E, and EXTRACT. (**e**) For scientific analysis, numerous general tools for exploration and visualization of NWB files (e.g. NWBWidgets and NWBExplorer) as well as application-specific tools for advanced analytics (e.g. Brainstorm) are accessible to the NWB community. (**f–g**) NWB is supported by a growing set of data archives (e.g. DANDI) for publication and preservation of research data. Data archives in conjunction with NWB APIs, validation tools, and the NDX Catalog play a central role in facilitating data reuse and discovery.

eventual publications when generated. The code repositories for the entire infrastructure are available on Github (**DANDI, 2022**) under an Apache 2.0 license.

## NWB is integrated with state-of-the-art analysis tools throughout the data life cycle

The goal of NWB is to accelerate the rate and improve the quality of scientific discovery through rigorous description and high-performance storage of neurophysiology data. Achieving this goal requires us to consider not just NWB, but the entire data life cycle, including planning, acquisition,

processing, and analysis to publication and reuse (*Figure 6*). NWB provides a common language for neurophysiology data collected using existing and emerging neurophysiology technologies integrated into a vibrant neurophysiology data ecosystem. We describe the software relating to NWB as an 'ecosystem', because it is a marketplace of a diverse set of tools each playing a different role, from data acquisition, to visualization, analysis, and publication. NWB allows scientists to identify an unmet need and contribute new tools to address this need. This is critical to truly make neurophysiology data Findable, Accessible, Interoperable, and Reusable (i.e. FAIR) (*Wilkinson et al., 2016*).

NWB supports experiment planning by helping users to clearly define what metadata to collect and how the data will be formatted and managed (*Figure 6a*). To support data acquisition, NWB allows for the storage of unprocessed acquired electrical and optical physiology signals. Storage of these signals requires either streaming the data directly to the NWB file from the acquisition system or converting data from other formats after acquisition (*Figure 6b and c*). Some acquisition systems, such as the MIES (*MIES, 2022*) intracellular electrophysiology acquisition platform, already support direct recording to NWB and the community is actively working to expand support for direct recording to NWB, for example, via ScanImage (*Pologruto et al., 2003*) and OpenEphys (*Siegle et al., 2017*). To allow utilization of legacy data and other acquisition systems, a variety of tools exist for converting neurophysiology data to NWB. For extracellular electrophysiology, the SpikeInterface package (*Buccino et al., 2019*) provides a uniform API for conversion and processing data that supports conversion for 19+different proprietary acquisition formats to NWB. For intracellular electrophysiology, the Intrinsic Physiology Feature Extractor (IPFX) package (*IPFX, 2021*) supports conversion of data acquired with Patchmaster. Direct conversion of raw data to NWB at the beginning of the data lifecycle facilitates data re-processing and re-analysis with up-to-date methods and data re-use more broadly.

The NWB community has been able to grow and integrate with an ecosystem of software tools that offer convenient methods for processing data from NWB files (and other formats) and writing the results into an NWB file (*Figure 6d*). NWB allows these tools to be easily accessed, compared, and used interoperably. Furthermore, storage of processed data in NWB files allows direct re-analysis of activation traces or spike times via novel analysis methods without having to reproduce time-intensive pre-processing steps. For example, the SpikeInterface API supports export of spike sorting results to NWB across nine different spike sorters and customizable data curation functions for interrogation of results from multiple spike sorters with common metrics. For optical physiology, several popular state-of-the-art software packages, such as CaImAn (*Giovannucci et al., 2019*), suite2p (*Pachitariu et al., 2016*), ciapkg (*Ahanonu, 2018*), and EXTRACT (*Inan et al., 2021*) help users build processing pipelines that segment optical images into regions of interest corresponding to putative neurons, and write these results to NWB.

There is also a range of general and application-specific tools emerging for analysis of neurophysiology data in NWB (*Figure 6e*). The NWBWidgets (*nwb-jupyter-widgets, 2022*) library enables interactive exploration of NWB files via web-based views of the NWB file hierarchy and dynamic plots of neural data, for example visualizations of spike trains and optical responses. NWB Explorer (*Cantarelli et al., 2022*) developed by MetaCell in collaboration with OpenSourceBrain, is a web app that allows a user to explore any publicly hosted NWB file and supports custom visualizations and analysis via Jupyter notebooks, as well as use of the NWBWidgets. These tools allow neuroscientists to inspect their own data for quality control, and enable data reusers to quickly understand the contents of a published NWB file. In addition to these general-purpose tools, many application-specific tools, for example, RAVE (*Magnotti et al., 2020*), ecogVIS (*Tauffer and Dichter, 2021*), Brainstorm (*Nasiotis et al., 2019*), Neo (*Garcia et al., 2014*) and others are already supporting or are in the process of developing support for analysis of NWB files.

Many journals and funding agencies are beginning to require that data be made FAIR. For publication and preservation (*Figure 6f*) archives are an essential component of the NWB ecosystem, allowing data producers to document data associated with publications and share that data with others. NWB files can be stored in many popular archives, such as FigShare and Collaborative Research in Computational Neuroscience (CRCNS.org). As described earlier, in the context of the NIH BRAIN Initiative, the DANDI archive has been specifically designed to publish and validate NWB files and leverage their structure for searching across datasets. In addition, several other archives, for example, DABI (*DABI, 2022*) and OpenSourceBrain (*Gleeson et al., 2019*), are also supporting publication of NWB data.

Data archives also play a crucial role in discovery and reuse of data (*Figure 6g*). In addition to providing core functionality for data storage and search, archives increasingly also provide compute capacity for reanalysis. For example, DANDI Hub provides users a familiar JupyterHub interface that supports interactive exploration and processing of NWB files stored in the archive. The NWB data APIs, validation, and inspection tools also play a critical role in data reuse by enabling access and ensuring data validity. Finally, the Neurodata Extension Catalog described earlier facilitates accessibility and reuse of data files that use NWB extensions.

NWB integrates with (not competes with) existing and emerging technologies across the data lifecycle, creating a flourishing NWB software ecosystem that enables users to access state-of-the-art analysis tools, share and reuse data, improve efficiency, reduce cost, and accelerate and enable the scientific discovery process. See also Appendix 4 for an overview of NWB-enabled tools organized by application area and environment. Thus, NWB provides a common language to describe neurophysiology experiments, data, and derived data products that enables users to maintain and exchange data throughout the data lifecycle and access state-of-the-art software tools.

| | FAIR Principles | Custom Binary Format | Zarr | HDF5 | NIX | NWB 1.0 | NWB 2.x | NWB + DANDI |
|---|---|---|---|---|---|---|---|---|
| **Findable** | F1. (Meta)data are assigned a globally unique and persistent identifier | | | | green | | green | green |
| | F2. Data are described with rich metadata (defined by R1 below) | | | | yellow | yellow | green | green |
| | F3. Metadata clearly and explicitly include the identifier of the data they describe | | | | yellow | yellow | green | green |
| | F4. (Meta)data are registered or indexed in a searchable resource | | | | | | | green |
| **Accessible** | A1. (Meta)data are retrievable by their identifier using a standardized communications protocol | | yellow | green | green | yellow | green | green |
| | A1.1 The protocol is open, free, and universally implementable | | yellow | green | green | green | green | green |
| | A1.2 The protocol allows for an authentication and authorization procedure, where necessary | | | | | | | green |
| | A2. Metadata are accessible, even when the data are no longer available | | | | | | | green |
| **Interoperable** | I1. (Meta)data use a formal, accessible, shared, and broadly applicable language for knowledge representation | | | | green | yellow | green | green |
| | I2. (Meta)data use vocabularies that follow FAIR principles | | | | yellow | yellow | green | green |
| | I3. (Meta)data include qualified references to other (meta)data | | | | yellow | yellow | green | green |
| **Reusable** | R1. (Meta)data are richly described with a plurality of accurate and relevant attributes | | | | yellow | green | green | green |
| | R1.1. (Meta)data are released with a clear and accessible data usage license | | | | | | | green |
| | R1.2. (Meta)data are associated with detailed provenance | | | | | green | green | green |
| | R1.3. (Meta)data meet domain-relevant community standards | | | | yellow | green | green | green |
| | **Summary Score** | 0 (+0) | 0 (+2) | 1 (+1) | 4 (+6) | 4 (+6) | 11 (+0) | 15 (+0) |

**Figure 7.** NWB together with DANDI provides an accessible approach for FAIR sharing of neurophysiology data. The table above assesses various approaches for sharing neurophysiology data with regard to their compliance with FAIR data principles. Here, cells shown in gray/green indicate non-compliance and compliance, respectively. Cells shown in yellow indicate partial compliance, either due to incomplete implementation or optional support, leaving achieving compliance ultimately to the end user. The larger, shaded blocks indicate areas that are typically not covered by data standards directly but are the role of other resources in a FAIR data ecosystem, e.g., the DANDI data archive.

## NWB and DANDI build the foundation for a FAIR neurophysiology data ecosystem

There have been previous efforts to standardize neurophysiology data, such as NWB(v1.0) and NIX (*Teeters et al., 2015*; *Martone et al., 2020*). While NWB(v1.0) drafted a standard for neurophysiology, it lacked generality which limited its scope, and did not have a reliable and rigorous software strategy and APIs, making it hard to use and unreliable in practice. In contrast, NIX defines a generic data model for storage of annotated scientific datasets, with APIs in C++ and Python and bindings for Java and MATLAB. As such, NIX provides important functionality towards building a FAIR data ecosystem. However, the NIX data model lacks specificity with regard to neurophysiology, leaving it up to the user to define appropriate schema to facilitate FAIR compliance. Due to this lack of specificity, NIX files can also be more varied in structure and naming conventions, which makes it difficult to aggregate across NIX datasets from different labs. In *Figure 7*, we assess and compare the compliance of different solutions for sharing neurophysiology data with FAIR data principles. The assessment for NIX is based on the INCF review for SPP endorsement (*Martone et al., 2020*). We also include a more in-depth breakdown of the assessment in Appendix 5. With increasing specificity of data models and standard schema—that is, as we move from general, self-describing formats (e.g. Zarr or HDF5) to generic data models (e.g. NIX) to application-specific standards (e.g. NWB)—compliance with FAIR principles and rigidness of the data specification increases. In practice, the various approaches often focus on different data challenges. As such, this is not an assessment of the quality of a product per-se, but an assessment of the out-of-the-box compliance with FAIR principles in the context of neurophysiology. For example, while self-describing data formats (like HDF5 or Zarr) lack specifics about (meta) data related to neurophysiology, they provide important technical solutions towards enabling high-performance data management and storage.

Complementary to standardization of data, software packages, e.g., Neo (*Garcia et al., 2014*), SpikeInterface (*Buccino et al., 2019*) and others, aim to simplify programmatic interaction with neurophysiology data in diverse formats and/or tools with diverse programming interfaces (e.g. for spike sorting) by providing common software interfaces for interacting with the data/tools. This strategy provides an important conduit to enable access to and facilitate integration with a diversity of data and tools. However, this approach does not address (nor does it aim to address) the issue of compliance of data with FAIR principles, but it rather aims to improve interoperability between and interaction with a diversity of tools and data formats. Ultimately, standardization of data and creation of common software interfaces are not competing strategies, but are synergistic approaches that together help create a more integrated data ecosystem. Indeed, tools such as SpikeInterface (*Buccino et al., 2019*), are an important component of the larger NWB software ecosystem that help create an accessible neurophysiology data ecosystem by making it easier for users to integrate their data and tools with NWB and facilitating access and interoperability of diverse tools.

Data standards build the foundation for an overall data strategy to ensure compliance with FAIR data principles. Ultimately, however, ensuring FAIR data sharing and use depends on an ecosystem of data standards and data management, analysis, visualization, and archive tools as well as laws, regulations, and data governance structures—for example the NIH BRAIN Initiative Data Sharing Policy (*NOT-MH-19-010, 2021*) or the OMB Open Data Policy (*Burwell et al., 2022*)—all working together (*Eke et al., 2022*). As *Figure 7* illustrates, it is ultimately the combination of NWB and DANDI working together that enable compliance with FAIR principles. Here, certain aspects, such as usage licenses (R.1.1), indexing and search (F.4), authenticated access (A1.2), and long-term availability of metadata (A2) are explicitly the role of the archive. As this table shows, together, NWB and DANDI make neurophysiology data FAIR.

## Coordinated community engagement, governance, and development of NWB

The neurophysiology community consists of a large diversity of stakeholders with vested interests and broad use cases. Inclusive engagement and outreach with the community are central to achieve acceptance and adoption of NWB and to ensure that NWB meets user needs. Thus, development of scientific data languages is as much a sociological challenge as it is a technological challenge. To address this challenge, NWB has adopted a modern open-source software strategy (*Figure 8a*) with community resources and governance, and a variety of engagement activities.

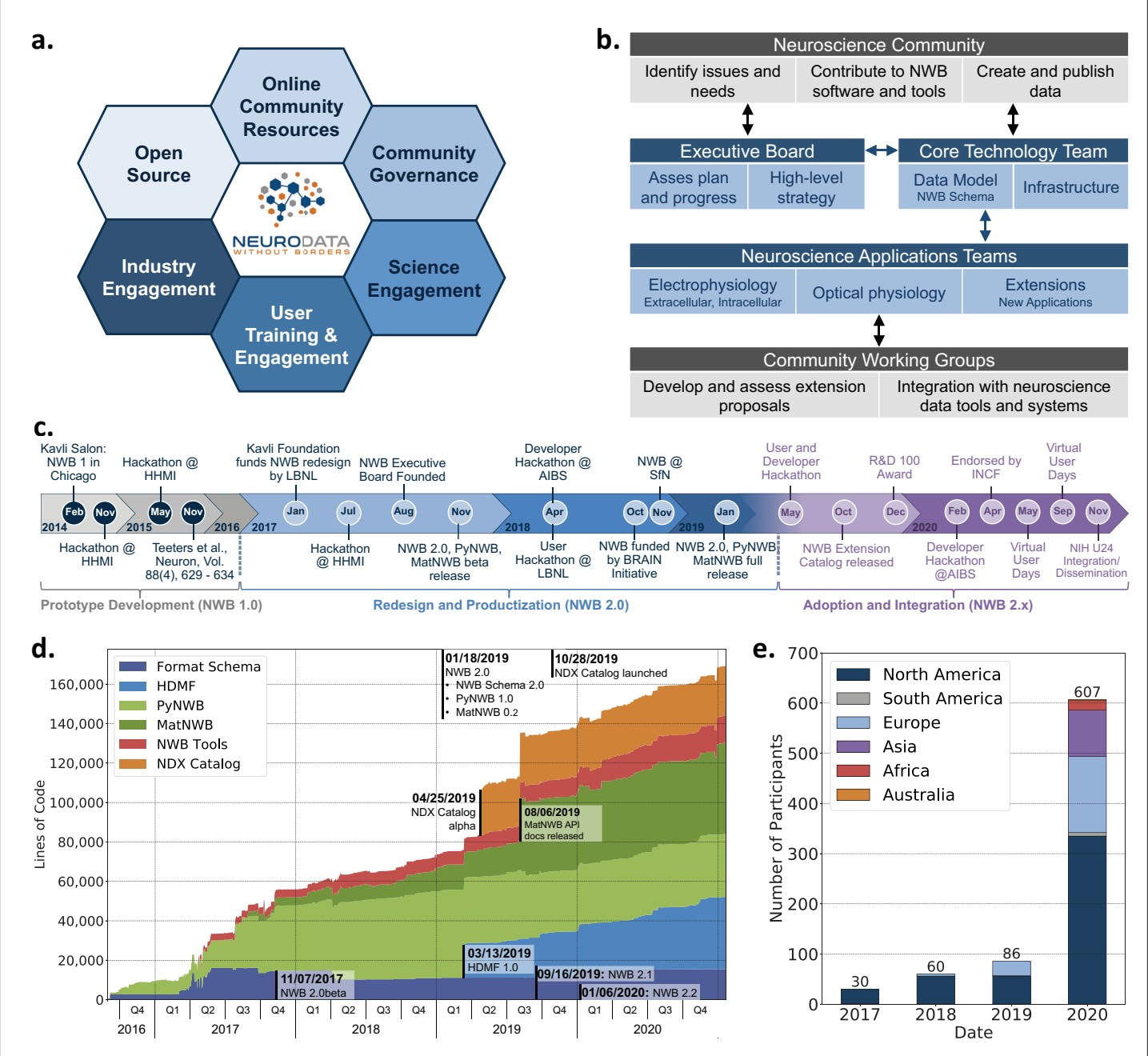

**Figure 8.** Coordinated community engagement, governance, and development of NWB. (**a**) NWB is open source with all software and documents available online via GitHub and the nwb.org website. NWB provides a broad range of online community resources, e.g., Slack, online Help Desk, GitHub, mailing list, or Twitter, to facilitate interaction with the community and provides a broad set of online documentation and tutorials. NWB uses an open governance model that is transparent to the community. Broad engagements with industry partners (e.g. DataJoint, Kitware, MathWorks, MBFBioscience, Vidrio, CatalystNeuro, etc.) and targeted science engagements with neuroscience labs and tool developers help sustain and grow the NWB ecosystem. Broad user training and engagement activities, e.g., via hackathons, virtual training, tutorials at conferences, or online training resources, aim at facilitating adoption and growing the NWB community knowledge base. (**b**) Organizational structure of NWB showing the main bodies of the NWB team (blue boxes) and the community (gray boxes), their roles (light blue/gray boxes), and typical interactions (arrows). (**c**) The timeline of the NWB project to date can be roughly divided into three main phases. The initial NWB pilot project (2014–2015) resulted in the creation of the first NWB 1.0 prototype data standard. The NWB 2.0 effort then focused on facilitating use and long-term sustainability by redesigning and productizing the data standard and developing a sustainable software strategy and governance structure for NWB (2017–2019). The release of NWB 2.0 in Jan. 2019 marked the beginning of the third main phase of the project, focused on adoption and integration of NWB with neuroscience tools and labs, maintenance, and continued evolution and refinement of the data standard. (**d**) Overview of the growth of core NWB 2.x software in lines of code over

*Figure 8 continued on next page*

*Figure 8 continued*

time. (**e**) Number of participants at NWB outreach and training events over time. In the count we considered only the NWB hackathons and User Days (see c.), the 2019 and 2020 NWB tutorial at Cosyne, and 2019 training at the OpenSourceBrain workshop (i.e. not including attendees at presentations at conferences, e.g., SfN).

Execution of this strategy requires coordinating efforts across stakeholders, use cases, and standard technologies to prioritize software development and resolve potential conflicts. Such coordination necessitates a governance structure reflecting the values of the project and the diverse composition of the community (*Figure 8b*). The NWB Executive Board consists of diverse experimental and computational neuroscientists from the community and serves as the steering committee for developing the long-term vision and strategy. The NWB Executive Board (see Acknowledgements) was established in 2007 as an independent body from the technical team and PIs of NWB grants. The NWB Core Technology Team leads and coordinates the development of the NWB data language and software infrastructure to ensure quality, stability, and consistency of NWB technologies, as well as timely response to user issues. The Core Technology Team reports regularly to and coordinates with the Executive Board. Neuroscience Application Teams, consisting of expert users and core developers lead, engagement with targeted neuroscience areas in electrophysiology, optical physiology, and other emerging applications. These application teams are responsible for developing extensions to the data standard, new features, and technology integration together with Community Working Groups. The working groups allow for an agile, community-driven development and evaluation of standard extensions and technologies, and allow users to directly engage with the evolution of NWB. The broader neuroscience community further contributes to NWB via issue tickets, contributions to NWB software and documentation, and by creating and publishing data in NWB. This governance and development structure emphasizes a balance between the stability of NWB technologies that ensure reliable production software, direct engagement with the community to ensure that NWB meets diverse stakeholder needs, and agile response to issues and emerging technologies.

The balance between stability, diversity, and agility is also reflected in the overall timeline of the NWB project (*Figure 8c*). The NWB 1.0 prototype focused on evaluation of existing technologies and community needs and development of a draft data standard. Building on this prototype, the NWB 2.0 project initially focused on the redesign and productization of NWB, emphasizing the creation of a sustainable software architecture, reliable data standard, and software ready for use. Following the first full release of NWB 2.0 in January 2019, the emphasis then shifted to adoption and integration of NWB. The goal has been to grow a community and software ecosystem as well as maintenance and continued refinement of NWB.

Together, these technical and community engagement efforts have resulted in a vibrant and growing ecosystem of public NWB data (Appendix 6) and tools utilizing NWB. The core NWB software stack has continued to grow steadily since the release of NWB 2.0 in January 2019, illustrating the need for continued development and maintenance of NWB (*Figure 8d*). See also Appendix 7 for an overview of the software release process and history for the NWB schema and APIs. In 2020 more than 600 scientists participated in NWB developer and user workshops and we have seen steady growth in attendance at NWB events over time (*Figure 8e*). At the same time, the global reach of NWB has also been increasing over time (*Figure 8e*). The NWB team also provides extensive online training resources, including video and code tutorials, detailed documentation, as well as guidelines and best practices (see and Materials and methods). Community liaisons provide expert consultation for labs adopting NWB and for creating customized data conversion software for individual labs. As the table in Appendix 6 shows, despite its young age relative to the neurophysiology community, NWB 2.0 is being adopted by a growing number of neuroscience laboratories and projects led by diverse principal investigators, creating a representative community where users can exchange and reuse data, with NWB as a common data language (*Chandravadia et al., 2020*).

## Discussion

Investigating the myriad functions of the brain across species necessitates a massive diversity and complexity of neurophysiology experiments. This diversity presents an outstanding barrier in meaningful sharing and collaborative analysis of the collected data, and ultimately prevents the data from

being FAIR. To overcome this barrier, we developed a data ecosystem based on the Neurodata Without Borders (NWB) data language and software. NWB is being utilized by more than 36 labs to enable unified storage and description of intracellular, extracellular, LFP, ECoG, and $Ca^{2+}$ data in fly, mice, rats, monkeys, humans, and simulations. To support the entire data lifecycle, NWB natively operates with processing, analysis, visualization, and data management tools, as exemplified by the ability to store both raw and pre-processed simultaneous electrophysiology and optophysiology data. Formal extension mechanisms enable NWB to co-evolve with the needs of the community. NWB enables DANDI to provide a data archive that also serves as a collaboration space for neurophysiology projects. Together, these technologies greatly enhance the FAIRness of neurophysiology data.

We argue that there are several key challenges that, until NWB, have not been successfully addressed and which ultimately hindered wide-spread adoption of a common standard by the diverse neurophysiology community. Conceptually, the complexity of the problem necessitates an interdisciplinary approach of neuroscientists, data and computer scientists, and scientific software engineers to identify and disentangle the components of the solution. Technologically, the software infrastructure instantiating the standard must integrate the separable components of user-facing interfaces (i.e. Application Programming Interfaces, APIs), data modeling, standard specification, data translation, and storage format. This must be done while maintaining sustainability, reliability, stability, and ease of use for the neurophysiologist. Furthermore, because science is advanced by both development of new acquisition techniques and experimental designs, mechanisms for extending the standard to unforeseen data and metadata are essential. Sociologically, the neuroscientific community must accept and adopt the standard, requiring coordinated community engagement, software development, and governance. NWB directly addresses these challenges.

## NWB as the *lingua franca* of neurophysiology data

Making neurophysiology data FAIR requires a paradigm shift in how we conceptualize the solution. Scientists need more than a rigid data format, but instead require a flexible data language. Such a language should enable scientists to communicate via data. Natural languages evolve with the concepts of the societies that use them, while still providing a stable basis that enables communication of common concepts. Similarly, a scientific data language should evolve with the scientific research community, and at the same time provide a standardized core that expresses common and established methods and data types. NWB is such a data language for neurophysiology experiments.

There are many parallels between NWB and natural languages as used today. The NWB specification language provides the basic tools and rules for creating the core concepts required to describe neurophysiology data, much like an alphabet and phonetic rules in natural language describe the creation of words. Likewise, the format schema provides the words and phrases (neurodata_types) of the data language and rules for how to compose them to form data documents (NWB files), much like a dictionary and grammatical rules for sentence and document structure. Similarly, flexible data storage methods allow NWB to manage and share data in different forms depending on the application, much like we store natural languages in many different mediums (e.g. via printed books, electronic records, or handwritten notes). User APIs (here HDMF, PyNWB, and MatNWB) provide the community with tools to create, read, and modify data documents and interact with core aspects of the language, similarly to text editors for natural language. NWB Extensions provide a mechanism to create, publish, and eventually integrate new modules into NWB to ensure it co-evolves with the tools and needs of the neurophysiology community, just as communities create new words to communicate emerging concepts. Finally, DANDI provides a cloud-based platform for archiving, sharing, and collaborative analysis of NWB data, much like a bookstore or Wikipedia. Together, these interacting components provide the basis of a data language and exchange medium neuroscience community that enables reproduction, interchange, and reuse of diverse neurophysiology data.

## NWB is community driven, professionally developed, and democratizes neurophysiology

Today, there are many data formats and tools used by the neuroscience community that are not interoperable. Often, formats and tools are specific to the lab and even the researcher. This level of specificity is a major impediment to sharing data and reproducing results, even within the same lab. More broadly, the resulting fragmentation of the data space reinforces siloed research, and

makes it difficult for datasets or software to be impactful on a community level. Our goal is for the NWB data language to be foundational in deepening collaborations between the community of neuroscientists. The current NWB software is the result of an intense, community-led, years-long collaboration between experimental and computational neuroscientists with data scientists and computer scientists. Core to the principles and success of NWB is to account for diverse perspectives and use cases in the development process, integration with community tools, and engage in community outreach and feedback collection. NWB is governed by a diverse group to ensure both the integrity of the software and that NWB continues to meet the needs of the neuroscience community.

As with all sophisticated scientific instruments, there is some training required to get a lab's data into NWB. Several training and outreach activities provide opportunities for the neuroscience community to learn how to most effectively utilize NWB. Tutorials, hackathons and user training events allow us to bring together neuroscientists who are passionate about open data and data management. These users bring their own data to convert or their own tools to integrate, which in turn makes the NWB community more diverse and representative of the overall neuroscience community. NWBs digital presence has accelerated during the COVID pandemic, and has allowed the community to grow internationally and at an exponential rate. Updates on Twitter and the website (https://www.nwb.org/), tutorials on YouTube, and free virtual hackathons, all are universally accessible and have helped achieve a global reach, interacting with scientists from countries that are too often left out. Together, these outreach activities combined with NWB and DANDI democratizes both neurophysiology data and analysis tools, as well as the extracted insights.

## The future of NWB

To address the next frontier in grand challenges associated with understanding the brain, the neuroscience community must integrate information across experiments spanning several orders of magnitude in spatial and temporal scales (*Bouchard et al., 2016*; *Bouchard et al., 2018*; *Bargmann, 2014*). This issue is particularly relevant in the current age of massive neuroscientific data sets generated by emerging technologies from the US BRAIN Initiative, Human Brain Project, and other brain research initiatives worldwide. Advanced data processing, machine learning, and artificial intelligence algorithms are required to make discoveries based on such massive volumes of data (*Sejnowski et al., 2014*; *Bouchard et al., 2016*; *Bouchard et al., 2018*). Currently, different domains of neuroscience (e.g. genomic/transcriptomic, anatomy, neurophysiology, etc.,) are supported by standards that are not coordinated. Building bridges across neuroscience domains will necessitate interaction between the standards, and will require substantial future efforts. There are nascent activities for compatibility between NWB and the Brain Imaging Data Structure (BIDS), for example, as part of the BIDS human intracranial neurophysiology ECoG/iEEG extension (*Holdgraf et al., 2019*), but further efforts in this and other areas are needed.

It is notoriously challenging to make neurophysiology data FAIR. Together, the NWB data language and the NWB-based DANDI data archive support a data science ecosystem for neurophysiology. NWB provides the underlying cohesion of this ecosystem through a common language for the description of data and experiments. However, like all languages, NWB must continue to adapt to accommodate advances in neuroscience technologies and the evolving community using that language. As adoption of NWB continues to grow, new needs and opportunities for further harmonization of metadata arise. A key ongoing focus area is on development and integration of ontologies with NWB to enhance specificity, accuracy, and interpretability of data fields. For example, there are NWB working groups on genotype and spatial coordinate representation, as well as the INCF Electrophysiology Stimulation Ontology working group (*Electrophysiology Stimulation Ontology Working Group, 2022*). Another key area is extending NWB to new areas, such as the ongoing working groups on integration of behavioral task descriptions with NWB (e.g. based on BEADL *Generator, 2022*) and enhanced integration of simulations with NWB. We strongly advocate for funding support of all aspects of the data-software life cycle (development, maintenance, integration, and distribution) to ensure the neuroscience community fully reaps the benefits of investment into neurophysiology tools and data acquisition.

## Core design principles and technologies for biological data languages

The problems addressed by NWB technologies are not unique to neurophysiology data. Indeed, as was recently discussed in *Powell, 2021*, lack of standards in genomics data is threatening the promise of that data. Many of the tools and concepts of the NWB data language can be applied to enhance standardization and exchange of data in biology more broadly. For example, the specification language, HDMF, the concept of extensions and the extension catalog are all general and broadly applicable technologies. Therefore, the impact of the methods and concepts we have described here has the potential to extend well beyond the boundaries of neurophysiology.

We developed design and implementation principles to create a robust, extensible, maintainable, and usable data ecosystem that embraces and enables FAIR data science across the breadth of neurophysiology data. Across biology, experimental diversity and data heterogeneity are the rule, not the exception (*Kandel et al., 2013*). Indeed, as biology faces the daunting frontier of understanding life from atoms to organisms, the complexity of experiments and multimodality of data will only increase. Therefore, the principles developed and deployed by NWB may provide a blueprint for creating data ecosystems across other fields of biology.

# Materials and methods
## NWB GitHub organizations

All NWB software is available open source via the following three GitHub organizations. The Neurodata Without Borders (*Neurodata Without Borders, 2022*) GitHub organization is used to manage all software resources related to core NWB software developed by the NWB developer community, for example, the PyNWB and MatNWB reference APIs. The HDMF development (*HDMF-c, 2022*) Github organization is used to publish all software related to the Hierarchical Data Modeling Framework (HDMF), including, HDMF, HDMF DocUtils, HDMF Common Schema and others. Finally, the NWB Extensions (*NDXCatalog-b, 2021*) GitHub organization is used to manage all software related to the NDX Catalog, including all extension registrations. Note, the catalog itself only stores metadata about NDXs, the source code of NDXs are often managed by the creators in dedicated repositories in their own organizations.

## HDMF software

HDMF software is available on GitHub using an open BSD licence model.

Hierarchical Data Modeling Framework (HDMF) is a Python package for working with hierarchical data. It provides APIs for specifying data models, reading and writing data to different storage backends, and representing data with Python objects. HDMF builds the foundation for the implementation of PyNWB and specification language. [Source] (*HDMF-a, 2021*) [Documentation] (*HDMF-b, 2022*) [Web] (*HDMF-dev, 2021*).

HDMF Documentation Utilities (hdmf-docutils) are a collection of utility tools for creating documentation for data standard schema defined using HDMF. The utilities support generation of reStructuredText (RST) documents directly from standard schema which can be compiled to a large variety of common document formats (e.g. HTML, PDF, epub, man and others) using Sphinx. [Source] (*hdmf-docutils, 2022*).

HDMF Common Schema defines a collection of common reusable data structures that build the foundation for modeling of advanced data formats, e.g., NWB. APIs for the HDMF common data types are implemented as part of the hdmf.common module in the HDMF library. [Source] (*hdmf-common-schema-a, 2022*) [Documentation] (*hdmf-common-schema-b, 2022*).

HDMF Schema Language provides an easy-to-use language for defining hierarchical data standards. APIs for creating and interacting with HDMF schema are implemented in HDMF. [Documentation] (*hdmf-schema-language, 2022*).

## NWB software

NWB software is available on GitHub using an open BSD licence model.

PyNWB is the Python reference API for NWB and provides a high-level interface for efficiently working with Neurodata stored in the NWB format. PyNWB is used by users to create and interact

with NWB and neuroscience tools to integrate with NWB. [Source] (**PyNWB-a, 2021**) [Documentation] (**PyNWB-b, 2021**).

MatNWB is the MATLAB reference API for NWB and provides an interface for efficiently working with Neurodata stored in the NWB format. MatNWB is used by both users and developers to create and interact with NWB and neuroscience tools to integrate with NWB. [Source] (**matnwb, 2021b**) [Documentation] (**matnwb, 2021a**).

NWBWidgets is an extensible library of widgets for visualizing NWB data in a Jupyter notebook (or lab). The widgets support navigation of the NWB file hierarchy and visualization of specific NWB data elements. [Source] (**nwb-jupyter-widgets, 2022**).

NWB Schema defines the complete NWB data standard specification. The schema is a collection of YAML files in the NWB specification language describing all neurodata_types supported by NWB and their organization in an NWB file. [Source] (**NWB Schema-a, 2022**) [Documentation] (**NWB Schema-b, 2021**).

NWB Schema Language is a specialized variant of the HDMF schema language. The language includes minor modifications (e.g., use of the term neurodata_type instead of data_type) to make the language more intuitive for neuroscience users, but it is otherwise identical to the HDMF schema language. Dedicated interfaces for creating and interacting with NWB schema are available in PyNWB. [Documentation] (**NWB Specification Language, 2022**).

NWB Storage defines the mapping of NWB specification language primitives to HDF5 for storage of NWB files. [Documentation] (**NWB Storage, 2021**).

Neurodata Extensions Catalog (NDX Catalog) is a community-led catalog of Neurodata Extensions (NDX) to the NWB data standard. All extensions mentioned in the text can be accessed directly via the catalog. [Source] (**NDXCatalog-b, 2021**) [Online] (**NDXCatalog-a, 2022**).

NWB Extensions Template (ndx-template) provides an easy-to-use template based on the Cookiecutter library for creating Neurodate Extensions (NDX) for the NWB data standard. [Source] (**NDXtemplate, 2022**).

NWB Staged Extensions is a repository for submitting new extensions to the NDX catalog [Source] (**staged extensions, 2021**).

## DANDI

The DANDI archive was created by developing and integrating several opensource projects and BRAIN Initiative data standards (NWB, BIDS, NIDM). The Web browser application is built using the VueJS framework and the DANDI command line interface is built using Python and PyNWB. The initial backend of the archive was built on top of the Girder data management system, and is transitioning to a Django-based framework. The DANDI analysis hub is built using Jupyterhub deployed over a Kubernetes cluster. The different components of the archive are hosted on Amazon Web Services and the Heroku platform. The code repositories for the entire infrastructure are available on Github (**DANDI, 2022**) under an Apache 2.0 license.

## Acknowledgements

Research reported in this publication was supported by the following grants and institutions. OR, AT, RL, and KEB were supported by the National Institute Of Mental Health of the National Institutes of Health under Award Number R24MH116922 (PI: O Ruebel) as well as through additional support by the Kavli Foundation (PI: KEB). BD and IS were supported by the BRAIN Initiative under award number U19-NS104590 to IS and by the Kavli Foundation. SG and BD were supported by the NIH BRAIN Initiative under award number 1R24MH117295-01A1. KS is supported by HHMI. Additional support for NWB was also provided by the Simons Foundation for the Global Brain grant 521,921 to LF, and with additional funding from the Kavli Foundation and Simons Foundation for MatNWB to KS. KEB was additionally supported by the Weill Neurohub. CatalystNeuro is also supported by the Simons Foundation.

We thank the diverse participants of the NWB hackathons and the global NWB user and developer community for their feedback and contributions. We thank Michael Grauer, Matt McCormick, Jean-Christophe Fillion-Robin, Doruk Ozturk, Roni Choudhury and Pamela Baker for early work on CI/CD. We thank the current and former members of the NWB Executive Board for their guidance and support: Kristofer Bouchard, Bing Brunton, Elizabeth Buffalo, Anne Churchland, Loren Frank, Satrajit

Ghosh, Adam Kepecs, Lydia Ng, Huib Mansvelder, Ueli Rutishauser, Karel Svoboda, Christof Koch, Friedrich Sommer, Markus Meister, and Katrin Amunts.

## Additional information

### Competing interests

Benjamin K Dichter: BD is the Founder and CEO of CatalystNeuro, a software consulting company that works with neurophysiology labs to build state-of-the-art data management workflows. Much of this work involves converting data from lab-specific formats to the NWB standard, and enhancing analysis and visualization tools to read and write NWB data. As such, Dr. Dichter has a personal financial state in the success of the NWB standard. Lawrence Niu: LN is a software engineer at MBF Bioscience, a for-profit biotech company that develops microscopy software and hardware. The other authors declare that no competing interests exist.

### Funding

| Funder | Grant reference number | Author |
|---|---|---|
| Kavli Foundation | | Oliver Rübel<br>Andrew Tritt<br>Ryan Ly<br>Benjamin K Dichter<br>Lawrence Niu<br>Ivan Soltesz<br>Karel Svoboda<br>Loren Frank<br>Kristofer E Bouchard |
| National Institute of Mental Health | R24MH116922 | Oliver Rübel<br>Andrew Tritt<br>Ryan Ly<br>Benjamin K Dichter<br>Kristofer E Bouchard<br>Pamela Baker<br>Lydia Ng |
| Howard Hughes Medical Institute | | Karel Svoboda<br>Loren Frank |
| Simmons Family Foundation | 521921 | Benjamin K Dichter<br>Lawrence Niu<br>Karel Svoboda<br>Loren Frank |
| National Institute of Mental Health | 1R24MH117295-01A1 | Benjamin K Dichter<br>Satrajit S Ghosh |
| National Institute of Neurological Disorders and Stroke | U19-NS104590 | Benjamin K Dichter<br>Ivan Soltesz |
| Weill Neurohub | | Kristofer E Bouchard |

The funders had no role in study design, data collection and interpretation, or the decision to submit the work for publication.

### Author contributions

Oliver Rübel, Conceptualization, Funding acquisition, Methodology, Project administration, Resources, Software, Supervision, Visualization, Writing – original draft, Writing – review and editing; Andrew Tritt, Ryan Ly, Methodology, Software, Visualization, Writing – original draft, Writing – review and editing; Benjamin K Dichter, Methodology, Visualization, Writing – original draft, Writing – review and editing; Satrajit Ghosh, Funding acquisition, Methodology, Visualization, Writing – original draft, Writing – review and editing; Lawrence Niu, Software; Pamela Baker, Software, Validation; Ivan Soltesz, Karel Svoboda, Loren Frank, Funding acquisition, Resources, Writing – review and editing; Lydia Ng, Conceptualization, Funding acquisition, Resources, Software, Supervision; Kristofer E Bouchard,

Conceptualization, Funding acquisition, Project administration, Resources, Supervision, Visualization, Writing – original draft, Writing – review and editing

**Author ORCIDs**
Oliver Rübel http://orcid.org/0000-0001-9902-1984
Ryan Ly http://orcid.org/0000-0001-9238-0642
Satrajit Ghosh http://orcid.org/0000-0002-5312-6729
Kristofer E Bouchard http://orcid.org/0000-0002-1974-4603

**Decision letter and Author response**
Decision letter https://doi.org/10.7554/eLife.78362.sa1
Author response https://doi.org/10.7554/eLife.78362.sa2

---

## Additional files

### Supplementary files
• MDAR checklist

### Data availability
All data used have been previously published and are available for download on the DANDI repository. All software is publicly available via GitHub: https://github.com/NeurodataWithoutBorders.

The following previously published datasets were used:

| Author(s) | Year | Dataset title | Dataset URL | Database and Identifier |
|---|---|---|---|---|
| Chung JE, Joo HR, Fan JL, Liu DF, Barnett AH, Chen S, Geaghan-Breiner C, Karlsson MP, Karlsson M, Lee KY, Liang H, Magland JF, Pebbles JA, Tooker AC, Greengard LF, Tolosa VM, Frank LM | 2019 | Polymer probe recordings from hippocampus (LFP), OFC, NAc, and mPFC | https://dandiarchive.org/dandiset/000065/draft | DANDI, 000065 |
| Bouchard KE, Mesgarani N, Johnson K, Chang EF | 2013 | Functional organization of human sensorimotor cortex for speech articulation | https://dandiarchive.org/dandiset/000019/0.220126.2148 | DANDI, 000019/0.220126.2148 |
| Raikov I, Milstein A, Soltesz I | 2022 | Simulation extension example | https://dandiarchive.org/dandiset/000064/0.221025.1735 | DANDI, 000064 |
| Huang L, Knoblich U, Ledochowitsch P, Lecoq J, Clay Reid R, de Vries SEJ, Buice MA, Murphy GJ, Waters J, Koch C, Zeng H, Li L | 2020 | Relationship between simultaneously recorded spiking activity and fluorescence signal in GCaMP6 transgenic mice | https://dandiarchive.org/dandiset/000048/draft | DANDI, 000048 |
| Gouwens NW, Sorensen SA, et al | 2020 | Cells Patch-seq recordings from mouse visual cortex; Integrated morphoelectric and transcriptomic classification of cortical GABAergic | https://dandiarchive.org/dandiset/000020 | DANDI, 000020 |

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

# Appendix 1

## Intracellular Electrophysiology Example using NWB and DANDI

The following shows a simple example, demonstrating the use of NWB for storage of intracellular electrophysiology data. The file used in this example is from DANDISET 20 (available at https://dandiarchive.org/dandiset/000020) from the Allen Institute for Brain Science as part of the multimodal characterization of cell types in the mouse visual cortex (see https://portal.brain-map.org/explore/classes/multimodal-characterization/multimodal-characterization-mouse-visual-cortex).

The following simple code example illustrates: (1) downloading of the file from DANDI, (2) reading the file with PyNWB, (3) visualization of the stimulus and response recording for a single sweep (*Appendix 1—figure 1*), and (4) visualization of the NWB file in NWB Widgets (*Appendix 1—figure 2*).

```python
# import required libraries
from dandi.dandiapi import DandiAPIClient
from pynwb import NWBHDF5IO
from nwbwidgets import nwb2widget
from nwbwidgets.timeseries import show_indexed_timeseries_mpl
import numpy as np
from matplotlib import pyplot as plt

# Determine the s3path for the file on DANDI
dandiset_id = '000020'
filepath = 'sub-1001658946/sub-1001658946_ses-1003020741_icephys.nwb'
with DandiAPIClient() as client:
    asset = client.get_dandiset(dandiset_id, 'draft').get_asset_by_path(filepath)
    s3_path = asset.get_content_url(follow_redirects=1, strip_query=True)
# Open the file using the ros3 driver for streaming data access
nwb_s3io =NWBHDF5IO(s3_path, mode='r', load_namespaces =True, driver='ros3')
# Read the file from DANDI. Here we only read the structure and
# attributes of the file, but not the bulk data
nwbfile =nwb_s3f.read()
# Create a simple example visualization of the response and stimulus
# timeseries for a single sweep
# get the timeseries associated with a particular sweep number
sweep_number =3
series =nwbfile.sweep_table.get_series(sweep_number)
# create a matplotlib figure for plotting
plt.rcParams['font.size'] = '16'
fig, (ax1, ax2)=plt.subplots(2, sharex =True, figsize=(12,8))
# plot the response and stimulus timeseries for the given sweep.
show_indexed_timeseries_mpl(series[0],
                            title =series[0].neurodata_type + ": "+series[0].name,
                            xlabel=None,
                            ax=ax1)
show_indexed_timeseries_mpl(series[1],
                            title=series(1).neurodata_type + ": "+series[1].name,
                            ax=ax2)

plt.show()
```

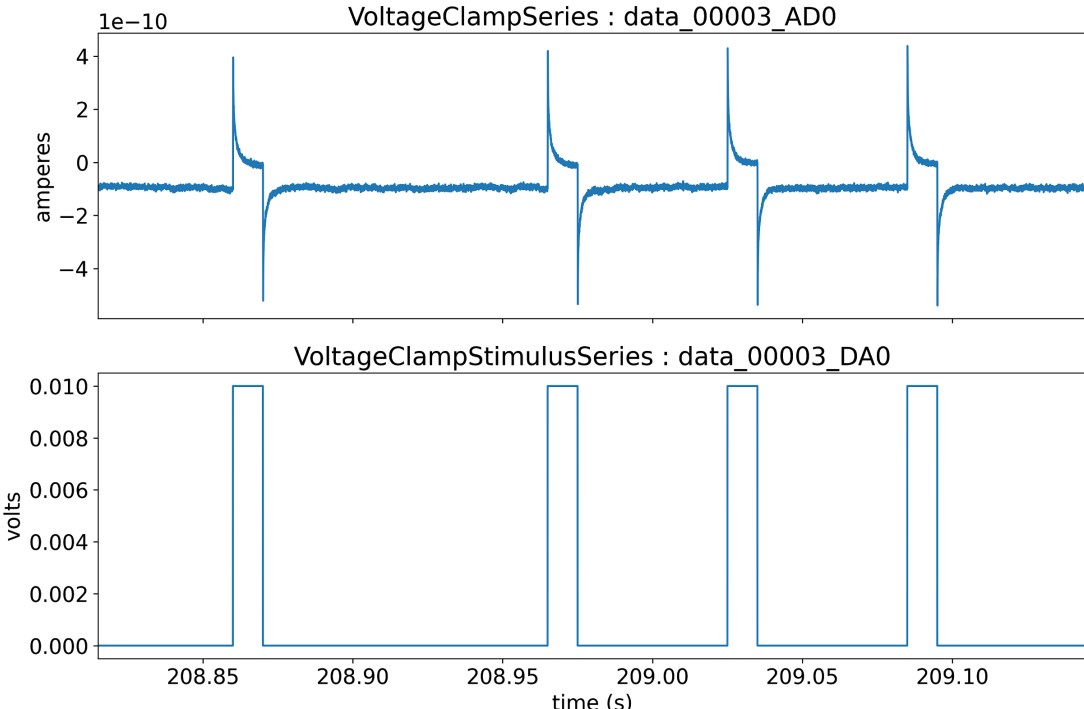

**Appendix 1—figure 1.** Visualization of the stimulus (bottom) and response (top) signals recorded via intracellular electrophysiology and stored in NWB.

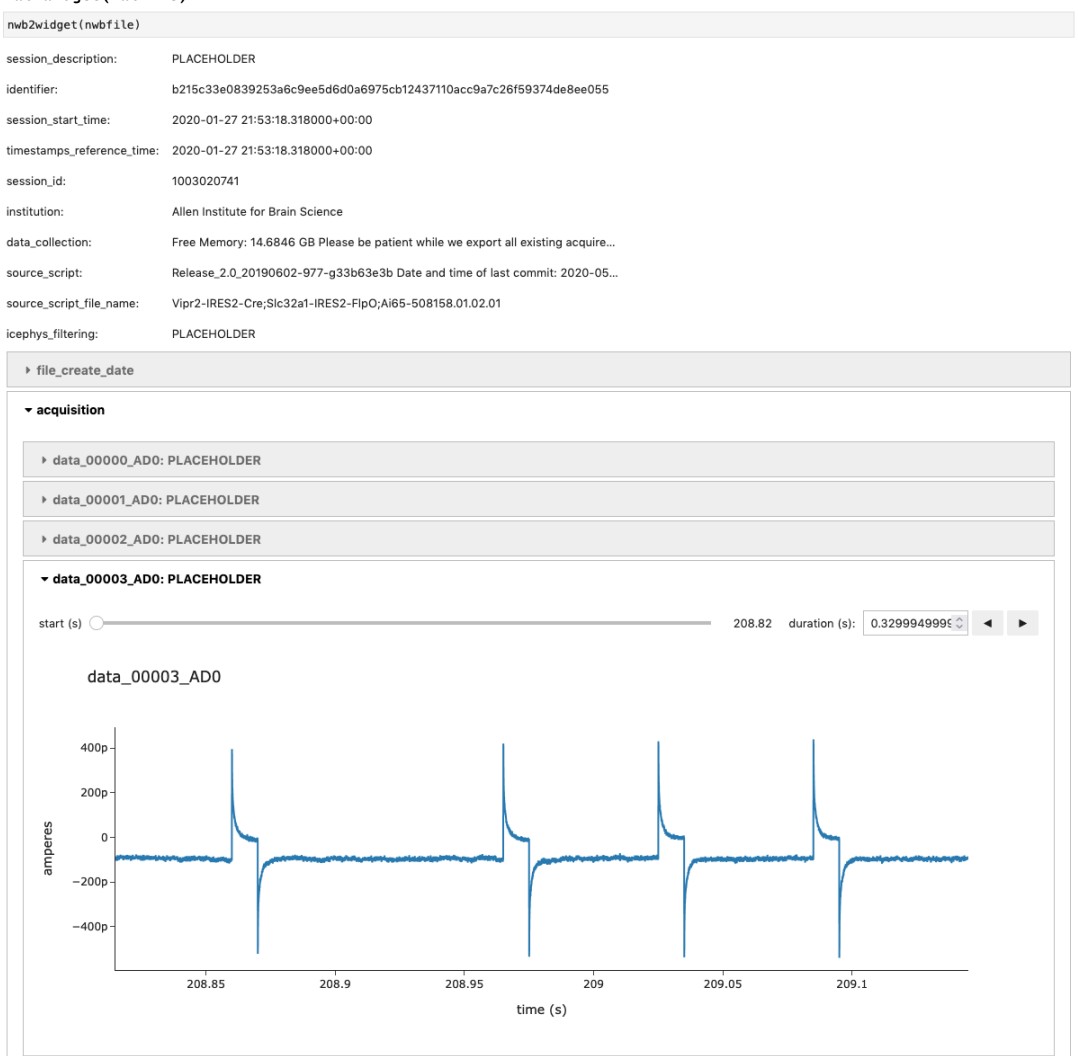

**Appendix 1—figure 2.** Visualization of the intracellular electrophysiology file using NWBWidgets.

## Appendix 2

### Creating a New Extension

Using the ndx-simulation-output extension (see *Figure 4c*) as an example, we illustrate in the following the main steps for creating a new extension outlined in *Figure 4a2*.

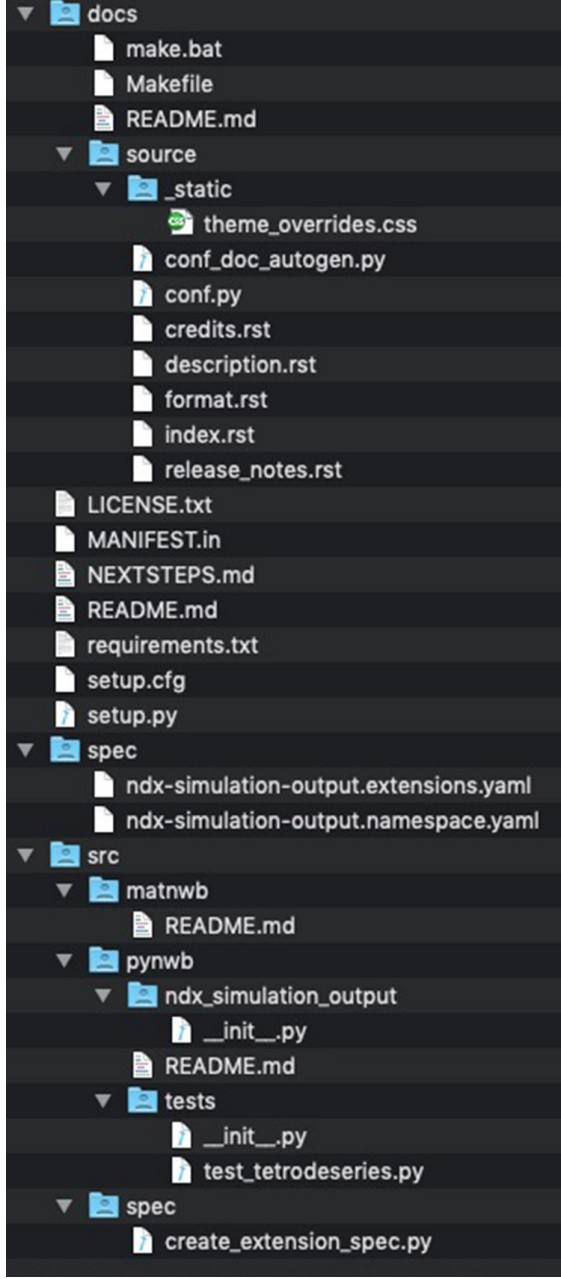

**Appendix 2—figure 1.** Files and folders generated by the cookiecutter ndx-template. The main folder contains the license and readme file for extension along with files required for installing the extension (e.g., *setup.py, setup.cfg, MANIFEST.in,* and *requirements.txt*) as well a markdown file with instructions for next steps. The *docs/* folder contains the Sphinx documentation setup for the extension. Without any additional changes required, the developer can with this setup automatically generate documentation in HTML, PDF, ePub and many other formats directly from the extension schema using the HDMF-DocUtils. Generating the documentation is as simple as executing "*make html*" in the *docs/* folder. The *spec/* folder contains the schema files for the extensions. The schema files are generated by the script in */src/spec/create_extension_spec.py* (see Define the Extension Schema

*Appendix 2—figure 1 continued on next page*

next), and are typically not modified manually by the developers. The */src* folder then contains main source codes for the extension, including the: *spec/* folder with the code to generated the extension schema *matnwb/* folder with code for MatNWB *pynwb/* folder with code for PyNWB.

## A2.1 Set up new NDX using the NDX template

The following code snippet shows the process for setting up the ndx-simulation-output extension using the ndx-template template. The template guides the developer through the setup via a simple question-and-answer process. In the code snippet, text shown in black is printed by cookiecutter, and text shown in blue are commands/responses entered by the developer.

```
>>cookiecutter gh:nwb-extensions/ndx-template
You've downloaded /Users/oruebel/.cookiecutters/ndx-template before. Is it okay to
delete and re-download it? [yes]: yes
namespace [ndx-my-namespace]: ndx-simulation-output
description [My NWB extension]: Data types for recording data from multiple
compartments of multiple neurons in a single TimeSeries.
author [My Name]: Ben Dichter
email [my_email@example.com]: ben.dichter@......
github_username [myname]: bendichter
copyright [2021, Ben Dichter]:
version [0.1.0]: 0.2.6
release [alpha]:
license [BSD 3-Clause]:
py_pkg_name [ndx_simulation_output]:
```

## A2.2 Define the extension schema

The code example below shows the */src/spec/create_extension_spec.py* script to define and generate the schema for the *ndx-simulation-output* extension using the PyNWB data format specification API. Code shown in red has been auto-generated by the *ndx-template*. Code shown in blue has been defined by the developer to create the schema. Running this script then automatically generates the YAML schema files for the extension stored in the *spec/* folder.

```
# -*- coding: utf-8 -*-
import os.path
from pynwb.spec import NWBNamespaceBuilder, export_spec, NWBGroupSpec
def main():
    # these arguments were auto-generated from your cookiecutter inputs
    ns_builder =NWBNamespaceBuilder(doc='Data types for recording data from multiple
compartments'
                                    'of multiple neurons in a single
TimeSeries.',
                                    name='ndx-simulation-output',
                                    version='0.2.6',
                                    author='Ben Dichter',
                                    contact='ben.dichter@gmail.com')
  types_to_include = ['TimeSeries', 'VectorData', 'VectorIndex', 'DynamicTable',
'LabMetaData']
  for ndtype in types_to_include:
      ns_builder.include_type(ndtype, namespace='core')
  Compartments = NWBGroupSpec(default_name='compartments',
                        neurodata_type_def='Compartments',
                        neurodata_type_inc='DynamicTable',
                      doc='Table that holds information about '
                          'what places are being recorded.')
  Compartments.add_dataset(name='number',
                      neurodata_type_inc='VectorData',
                      dtype='int',
                      doc='Cell compartment ids corresponding to a each column in the
data.')
      Compartments.add_dataset(name='number_index',
                      neurodata_type_inc='VectorIndex',
```

```
                              doc='Index that maps cell to compartments.',
                              quantity='?')
          Compartments.add_dataset(name='position',
                              neurodata_type_inc='VectorData',
                              dtype='float',
                              quantity='?',
                              doc='Position of recording within a compartment. '
                                  '0 is close to soma, 1 is other end.')
          Compartments.add_dataset(name='position_index',
                              neurodata_type_inc='VectorIndex',
                              doc='Index for position.',
                              quantity='?')
          Compartments.add_dataset(name='label',
                              neurodata_type_inc='VectorData',
                              doc='Labels for compartments.',
                              dtype='text',
                              quantity='?')
          Compartments.add_dataset(name='label_index',
                              neurodata_type_inc='VectorIndex',
                              doc='indexes label',
                              quantity='?')
          CompartmentsSeries = NWBGroupSpec(neurodata_type_def='CompartmentSeries',
                              neurodata_type_inc='TimeSeries',
                              doc='Stores continuous data from cell compartments')
          CompartmentsSeries.add_link(name='compartments',
                              target_type='Compartments',
                              quantity='?',
                              doc='Metadata about compartments in this CompartmentSeries.')
          SimulationMetaData = NWBGroupSpec(name='simulation',
                              neurodata_type_def='SimulationMetaData',
                              neurodata_type_inc='LabMetaData',
                              doc='Group that holds metadata for simulations.')
          SimulationMetaData.add_group(name='compartments',
                              neurodata_type_inc='Compartments',
                              doc='Table that holds information about '
                                  'what places are being recorded.')
          new_data_types = [Compartments, CompartmentsSeries, SimulationMetaData]
          # export the spec to yaml files in the spec folder
          output_dir =os.path.abspath(os.path.join(os.path.dirname(__file__), '..', '..',
      'spec'))
          export_spec(ns_builder, new_data_types, output_dir)
      if __name__ == "__main__":
      main()
```

## A2.3 Create API classes

The following code example is an abbreviated version of the *simulation_output.py* file located at *ndx-simulation-output/src/pynwb/ndx_simulation_output/* as part of the *ndx-simulation-output* extensions. For illustration purposes and to allow us to focus on the code relevant to the definition of the API classes, we here omit code details of the *find_compartments* function, which defines custom functionality.

The example shown here illustrates three main patterns for creating API classes for extensions. In part A. the extension uses the *get_class* method to dynamically generate an API Container class for the *SimulationMetaData* type directly from the schema. In part B. the extension uses the same approach for *CompartmentSeries*, but then further customizes the class by adding the *find_compartments* to the class to provide additional user functionality. In part C. the extension then defines a custom API Container class for the *Compartments* type that extends the *DynamicTable* type.

```
# Import methods for registering and creating container class
from pynwb import register_class, docval, get_class
# Import the docval decorator used for documenting functions and type checking
from hdmf.utils import docval, call_docval_func
# Import the base Container classes we are extending
from hdmf.common.table import DynamicTable, ElementIdentifiers
# Define the name of the namespace of our extension needed to register Container
```

```
classes

namespace = 'ndx-simulation-output'

# A. Auto-generate a Container class for the SimuluationMetaData type

SimulationMetaData = get_class('SimulationMetaData', namespace)

# B. Auto-generate a Container class for the CompartmentSeries type
CompartmentSeries =get_class('CompartmentSeries', namespace)
# B.1. Use monkey patching to add custom functionality to the auto-generated class
def find_compartments(self, cell, compartment_numbers =None, compartment_labels
=None):
  [...] # Details of the find_compartment omitted here for clarity.
CompartmentSeries.find_compartments =find_compartments

# C. Define a custom Container class for the Compartments table type
@register_class('Compartments', namespace) # Register the class with the TypeMap
class Compartments(DynamicTable):
    # Define the columns for the table. HDMF then automatically handles
    # setting up the columns for us as part of the class
    __columns__ = (
        {'name': 'number','index': True,
        'description': 'cell compartment ids corresponding to a each column in the
data'},
        {'name': 'position','index': True,
        'description': 'the observation intervals for each unit'},
        {'name': 'label','description': 'the electrodes that each spike unit came
from','index': True, 'table': True}
 )
   # Document and define the allowable types for the parameters of the __init__
function
    @docval({'name': 'name','type': str,
            'doc': 'Name of this Compartments object','default': 'compartments'},
            {'name': 'id','type': ('array_data', ElementIdentifiers),
            'doc': 'the identifiers for the units stored in this
interface','default': None},
            {'name': 'columns','type': (tuple, list),
            'doc': 'the columns in this table','default': None},
            {'name': 'colnames','type': 'array_data',
            'doc': 'the names of the columns in this table','default': None},
            {'name': 'description','type': str,
            'doc': 'a description of what is in this table',
            'default': 'Table that holds information about what places are being
recorded.'},
)
    def __init__(self, **kwargs):
        call_docval_func(super(Compartments, self).__init__, kwargs)
```

## A2.4 Documenting the extension

The *ndx-template* automatically generates as part of the *docs/* folder the full setup for automatically generating Sphinx-based documentation for the extension from the schema using the hdmf-docutils library. To generate the documentation we simply need to run the command *"make html"* in the *docs/* folder. Using the same approach we can generate documentation in a large range of common formats, e.g., HTML, PDF, man, or ePub. The ndx-template also generates standard *credits.rst*, *format.rst*, *release_notes.rst*, *description.rst*, and *index.rst* source files to make it easy for developers to customize the documentation and include additional details about the extension.

## Appendix 3

## Process for Creating, Publishing, and Updating Neurodata Extensions (NDX)

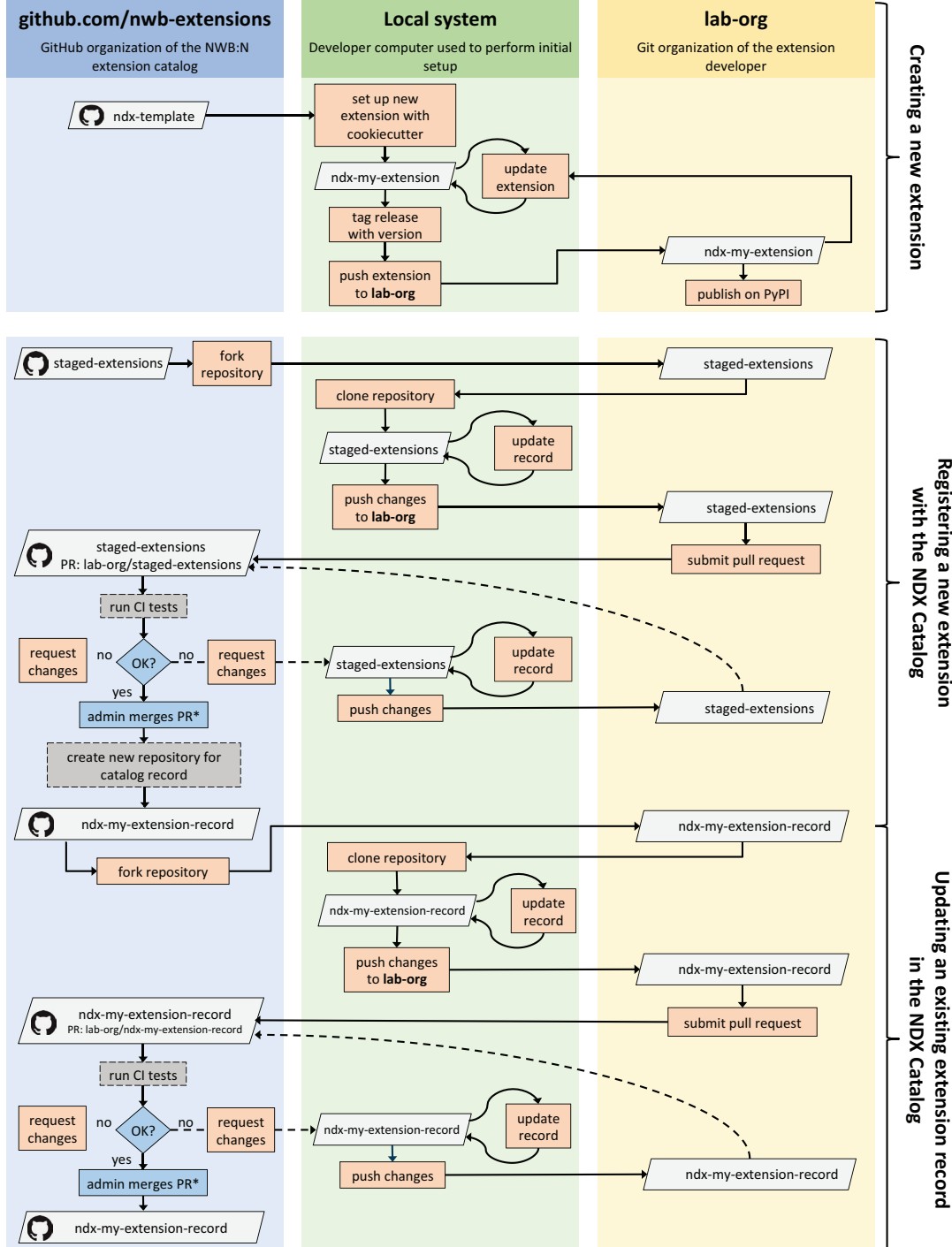

**Appendix 3—figure 1.** Illustration of the process for creating, publishing, and updating extensions via the Neurodata Extension Catalog (NDX Catalog), and (3) updating an extension/record. Boxes shown in gray indicate Git repositories; boxes in orange describe user actions; and boxes in blue indicate actions by administrators of the NDX catalog.

*Appendix 3—figure 1* shows an overview of the process for (1) creating a new extension, (2) creating a record to publish an extension via the Neurodata Extension Catalog (NDX Catalog), and (3) updating an extension/record. The figure also illustrates the automated CI processes that are managed in the NDX Catalog. The catalog process is modeled after the conda-forge model, which enables automation of many catalog processes using free, public services and avoiding the need for NWB to host its own services.

In the NDX Catalog, extensions are shared via the dedicated *nwb-extensions* GitHub organization for the NDX Catalog (blue column). The NDX Catalog provides the *ndx-template* cookiecutter extension template repository as well as the *staged-extensions* repository for submitting extensions to the NDX Catalog. Each extension record is then managed in a corresponding ndx record Git repository as part the *nwb-extensions* GitHub organization. The public NDX record repositories contains a *README.md* file describing the extension along with a *ndx-meta.yaml* metadata record for extensions, with basic information required for installing and locating the extension (see *Appendix 3—figure 2*).

Creation and changes to the extension on record are usually performed by a developer on their local system, e.g., laptop computer (green column). The developer then submits the changes to the extension or record repository via pull requests.

In this process, the Git repository with the sources of the extension remains in the lab organization of the submitter (yellow column). Here the only requirements are that: (1) the extension is stored in a Git repository and (2) the repository must be publicly accessible via the organization of the submitter, such that the repository can be cloned directly from the source location indicated in the NDX metadata record (*Appendix 3—figure 2*). This strategy allows for labs, universities, and independent groups to maintain ownership of the source code for their extensions in their own public Git space (e.g, on GitHub, Bitbucket, or GitLab) while creating an open, standardized record of all public extensions in a central location as part of the NDX Catalog. The ability for submitters to retain ownership of their extensions in their own organization is important to facilitate development as well as to retain a clear chain of responsibility and ownership. This is particularly important when the developers of the extension are funded by their own grants and/or are applying for funding.

```
name: ndx-simulation-output
version: 0.2.6
src: https://github.com/bendichter/ndx-simulation-output
pip: https://pypi.org/project/ndx-simulation-output
license: BSD
maintainers:
  - bendichter
```

**Appendix 3—figure 2.** Example *ndx-meta.yaml* metadata record for the *ndx-simulation-output* extension.

## Appendix 4

## Overview of select data analysis, visualization, and management tools that support NWB

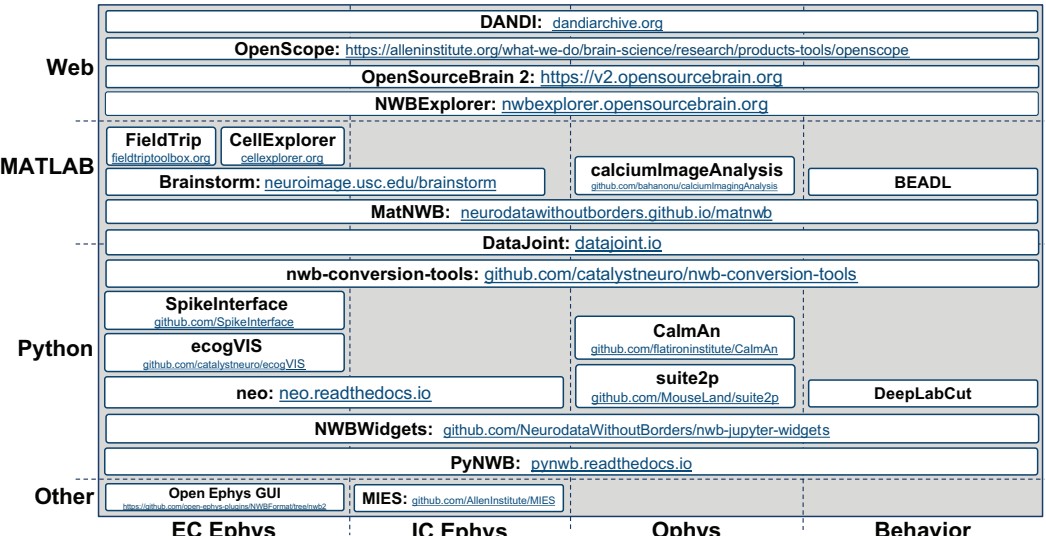

**Appendix 4—figure 1.** Overview of select data analysis, visualization, and management tools that support NWB. Visualization showing select data analysis, visualization, and management tools that support NWB organized by their main application (x-axis) and programming environment (y-axis).

# Appendix 5

## Assessment of FAIRness of NWB + DANDI

*Appendix 5—tables 1–4* below assess different solutions for sharing neurophysiology data with regard to their compliance with FAIR data principles, with cells shown in: (i) gray indicate non-compliance, (ii) green indicate compliance, and (iii) yellow indicate partial compliance either due to incomplete implementation or optional support, leaving achieving compliance ultimately to the end user. The assessment for NIX is based on the INCF review for SPP endorsement (https://doi.org/10.7490/f1000research.1117858.1). The "Custom" row in the tables refers to lab-specific binary formats.

In practice, the various approaches target different principle uses, and as such this is not an assessment of the quality of a product per-se, but rather its out-of-the-box compliance with FAIR principles in the context of neurophysiology. For example, self-describing data formats (like HDF5 or Zarr) seek to address challenges in high-performance data management and storage independent of a particular application, and as such lack specifics about (meta)data related to neurophysiology. However, while self-describing formats (like HDF5) are by themselves not sufficient to achieve FAIR compliance, they still form a critical building block in an overall strategy for FAIR data as evidenced by the fact that NIX, NWB, and many other application standards across science domains build on HDF5. Similarly, NIX provides a generic data model to enable storage of "fully annotated scientific datasets, i.e. the data together with its metadata within the same container" with the goal to enable "standardization by providing a common/generic data structure for a multitude of data types" (text in italic quoted from http://g-node.github.io/nix/). As such, NIX provides important functionality towards building a FAIR data strategy, but the NIX data model by itself lacks specificity with regard to neurophysiology, leaving it up to the user to define appropriate schema to facilitate FAIR compliance. Broadly speaking, with increasing specificity of data standards——i.e., as we move from general-purpose, self-describing formats (Zarr, HDF5) to generic data standards (NIX) to application-specific standards (NWB)— compliance with FAIR principles and rigidness of the data specification increases.

**Appendix 5—table 1.** Compliance of NWB+DANDI with FAIR principles: Findability.

| | Findable | | | |
|---|---|---|---|---|
| | F1. (Meta)data are assigned a globally unique and persistent identifier | F2. Data are described with rich metadata (defined by R1 below) | F3. Metadata clearly and explicitly include the identifier of the data they describe | F4. (Meta)data are registered or indexed in a searchable resource |
| Custom | No | No | No | |
| Zarr | No | • Self-describing, structural metadata (e.g., data type, array shape etc.) only | | |
| HDF5 | No | • Scientific (meta)data is fully user defined | | |
| NIX | • UUIDs are assigned to all objects | • Self-describing, structural metadata (uses HDF5)<br>• Generic data model (i.e., scientific (meta)data is user-defined) | | |
| NWB 1.0 | No | • Yes, but the schema language was not formally defined | • Similar to NWB 2 .x but the much more flexible schema (including inclusion of arbitrary data) often lead to non-compliance | • N/A. This is a key function of data archives and management systems |
| NWB 2 .x | • UUIDs are assigned to all objects<br>• External file identifier can be stored in the *identifier* field | • Rich schema for neurophysiology (meta)data<br>• Self-describing, structural metadata (uses HDF5) constrained by the standard schema | • Metadata is either directly associated with or explicitly linked to by the corresponding objects | |
| DANDI | • All dandisets and assets carry unique and persistent identifiers | • Uses NWB and other modern data standards<br>• Provides its own Dandiset schema for metadata about whole data collections | • Yes, persistent identifiers used by the archive are included with the metadata | • DANDI is a public archive that features rich search features over publicly shared data |

**Appendix 5—table 2.** Compliance of NWB+DANDI with FAIR principles: Accessibility.

| | Accessible | | | |
|---|---|---|---|---|
| | A1. (Meta)data are retrievable by their identifier using a standardised communications protocol | A1.1 The protocol is open, free, and universally implementable | A1.2 The protocol allows for an authentication and authorisation procedure, where necessary | A2. Metadata are accessible, even when the data are no longer available |
| Custom | No | No | | |
| Zarr | • Non-persistent file/object paths only | • Yes, but python-only API<br>• Long-term support is not clear | | |
| HDF5 | | • Portable format with broad support across programming languages and compute systems<br>• Intended for long-term support | | • N/A. This is a key function of data archives and management systems |
| NIX | • Yes | • Uses HDF5<br>• NIX API for C++. Matlab, Python and Java<br>• Open source | • N/A. This is a key function of data archives and management systems<br>• Encryption of files is possible via external tools<br>• HDF5/Zarr could support encryption of data elements via I/O filters | |
| NWB 1.0 | • Non-persistent file/object paths only (same as HDF5) | • Yes, but schema language was not formally defined and available APIs were limited | | |
| NWB 2 .x | • Yes. Objects retrievable based on UUID and path. | • Uses HDF5<br>• NWB API in Python and Matlab<br>• Open source | | |
| DANDI | • Uses NWB<br>• Metadata is exported as JSON/JSON-LD alongside with data<br>• REST API, Python, CLI, DataLad, ROS3 HDF5 | • Uses standard protocols (e.g., REST API)<br>• Supports integration with external services | • Supports user authentication and authorized access to all Dandisets, assets and other DANDI resources | • Searchable on the the archive and exposed as LinkedData |

**Appendix 5—table 3.** Compliance of NWB+DANDI with FAIR principles: Interoperability.

| | Interoperable | | |
|---|---|---|---|
| | I1. (Meta)data uses a formal, accessible, shared, and broadly applicable language for knowledge representation. | I2. (Meta)data use vocabularies that follow FAIR principles | I3. (Meta)data include qualified references to other (meta)data |
| Custom | No | No | No |
| Zarr | No | No | No |
| HDF5 | No | No | No |
| NIX | • Uses odML<br>• Uses HDF5 | • User defined | • User defined |
| NWB 1.0 | • Uses custom schema definition in Python | • Data follows the NWB 1.0 schema | • Partially. NWB 2 .x significantly enhanced support for linking of metadata with data. |
| NWB 2 .x | • Schema defined in JSON/YAML using json-schema<br>• NWB and extension schema are available with NWB files and online<br>• Uses HDF5 | • Data follows the NWB schema<br>• NWB supports use of ontologies via linking to external resources* | • The NWB schema explicitly models links between (meta)data<br>• NWB supports linking to external resources[3] |
| DANDI | • Uses NWB, JSON +json-schema, JSON-LD | • Uses NWB and other FAIR ontologies | • schema.org, spdx.org (licenses), PROV |

*Support for external resources has been released in HDMF >2.3 and is currently undergoing community review for integration with the NWB core data standard.

**Appendix 5—table 4.** Compliance of NWB+DANDI with FAIR principles: Reusability.

| | Reusable | | | |
| --- | --- | --- | --- | --- |
| | R1. (Meta)data are richly described with a plurality of accurate and relevant attributes | R1.1. (Meta)data are released with a clear and accessible data usage license | R1.2. (Meta)data are associated with detailed provenance | R1.3. (Meta)data meet domain-relevant community standards |
| Custom | No | | No | No |
| Zarr | No | | No | No |
| HDF5 | No | | No | No |
| NIX | • User defined | • N/A. Usage licences are typically managed by data archives | No | • User defined |
| NWB 1.0 | • Yes | | • Yes. NWB 2 .x further refined this significantly | • Yes |
| NWB 2 .x | • Yes | | • Includes detailed metadata about publications, experimenters, devices, subjects etc.<br>• Derived data (e.g., ROIs) link to the source data | • Yes, NWB provides detailed, neurophysiology-specific data schema |
| DANDI | • Uses NWB and defined dandiset schema | • All data in DANDI is published with a clear data usage licence | • Dandisets support detailed metadata about the data generation<br>• Dandisets are versioned | • Uses NWB |

## Appendix 6

The table shown below provides an overview of select labs that are using NWB. The last column of the table lists relevant DANDI datasets that have been published via the DANDI data archive using NWB. Each DANDI dataset typically consists of a large collection of NWB files related to a particular publication or experiment, with each NWB file representing the data from a particular recording session. All DANDI datasets can be found online at https://dandiarchive.org/dandiset/{6-digit-zero-padded-id}, e.g., https://dandiarchive.org/dandiset/000007. In the "Modality" column of the table we use the following abbreviations:

- ecephys: extracellular electrophysiology
- icephys: intracellular electrophysiology
- ophys: optical physiology
- ECoG: Electrocorticography
- fNIRS: Functional near-infrared spectroscopy

| Name, Affiliation | Species | Modality | DANDI datasets |
|---|---|---|---|
| AE Studio | human | fNIRS | 122 |
| Allen Institute | mouse, human | ecephys, icephys, ophys | 12, 20, 21, 22, 23, 24, 30, 36, 37, 39, 42, 43, 48, 49, 50, 66, 107, 109, 142, 209 |
| R. Axel, Columbia | fly | ophys | |
| Blue Brain Project | mouse | icephys | 25 |
| J. Berke, UCSF | rat | ecephys | |
| K.Bouchard, LBNL/UC Berkeley | rat, simulation | ecephys, uECoG | |
| C. Brody, Princeton | rat, mouse | ecephys | |
| B. Brunton, U Washington | human | ECoG | 55 |
| E. Buffalo, U Washington | monkey | ecephys | |
| T. Buschman, Princeton | monkey | ecephys | |
| G. Buzsaki, NYU | rat, mouse | ecephys | 3, 41, 44, 56, 59, 61, 67, 166, 213, 218 |
| M. Capogna, Aarhus | mouse | ecephys | |
| M. Carandini, UCL | mouse | ecephys | 17 |
| E. Chang, UCSF | human | ECoG | 19 |
| A. Churchland, CSHL | mouse | ecephys, ophys | 16 |
| R. Cossart, Inserm | mouse | ophys | 219 |
| D. Feldman, UC Berkeley | mouse | ecephys | |
| A. Fleischmann, Brown | mouse | ophys | 167 |
| L. Frank, UCSF | mouse | ecephys | 65, 115, 165 |
| L. Giocomo, Stanford | mouse | ecephys, ophys | 53, 54 |
| A. Groh, Heidelberg | mouse | ecephys, ophys | |
| K. Harris, UCL | mouse | ecephys | 17 |
| M. Hennig, Edinburgh | mouse | ecephys | 28, 34 |
| S. Husainni, Columbia | mouse | ecephys | |
| International Brain Lab | mouse | ecephys | 45, 149 |
| M. Jazayeri, MIT | monkey | ecephys | 130 |
| D. Jaeger, Emory | mouse | ophys, ecephys, icephys | |
| S. Kastner, Princeton | monkey | ecephys | |
| N. Li, Baylor | mouse | ecephys | 7 |
| A. Losonczy, Columbia | mouse | ophys | |
| G. Maimon, Rockefeller | fly | behavior | 212 |
| J. Martinez, Western | mouse, monkey, human | icephys | |

*Continued on next page*

*Continued*

| Name, Affiliation | Species | Modality | DANDI datasets |
|---|---|---|---|
| R. McGreal, UCSB | mouse | ophys | 206 |
| L. Miller, Northwestern | monkey | ecephys | 127 |
| T. Movshon, NYU | monkey | ecephys | |
| D. O'Connor, Johns Hopkins | mouse | ecephys, ophys | |
| J. Parvizi, Stanford | human | ECoG | |
| U. Rutishauser, Cedars-Sinai | human | ecephys | 4, 207 |
| B. Sabatini, Harvard | mouse | icephys | |
| S. Schultz, Imperial | mouse | ecephys, ophys | |
| K. Shenoy, Stanford | monkey | ecephys | 70, 121 |
| M. Smear, U Oregon | mouse | behavior | 217 |
| S. Smith, UCSB | mouse | ophys | 206 |
| I. Soltesz, Stanford | mouse, simulation | ophys, icephys | |
| N. Steinmetz, U Washington | mouse | ecephys | 17 |
| K. Svoboda, Janelia | mouse | ecephys, ophys | 5, 6, 9, 10, 11, 13, 15, 60, 168 |
| N. Tandon, UT Houston | human | ECoG | |
| D. Tank, Princeton | mouse | ecephys | |
| H. Tao, USC | mouse | icephys | 117 |
| A. Tolias, Baylor | mouse | icephys | 8, 35 |
| S. Tripathy, UofToronto/CAMH | human, mouse | icephys | |
| T. Valiante, Toronto | human | icephys | |

## Appendix 7

## Software Release Process and History

Software releases and processes are one indicator for the maturity of software products. As such, looking at the release history of NWB also provides some insight into how NWB has evolved over the course of the project from a first prototype of NWB 2 to a production data standard and software ecosystem.

The initial development of NWB 2 occurred during Nov. 2016 – Nov.2017. This phase did not include formal releases as the focus was on agile and rapid development of functionality with the goal to establish design principles and create a usable, fully functional prototype. Changes to the standard and software were evaluated in this phase by early adopters and reviewed by the community as part of NWB community hackathons.

The first beta release of the NWB 2 schema, PyNWB 0.2.0, and MatNWB 0.1.0b then occurred in November 2017 in conjunction with SfN. This marked the start of the beta testing and development phase of NWB, which occurred between Nov.2017 – Jan.2019. During the beta phase NWB adopted a more formal release process of versioned releases via pip, conda, and GitHub. These releases were targeted at early adopters and beta testers while development was still largely agile based on GitHub source releases. During this phase the focus was on evaluation, refinement, and productization.

In Jan.2019 we then released the first official version of NWB 2, including nwb-schema 2.0, PyNWB 1.0, and matnwb 0.2.0. This marked the beginning of the adoption and integration phase for the NWB 2 project. During Jan.2019 – Apr. 2021, a key focus has been on the one hand to continue to advance and refine NWB to meet the needs of adopters as well as to work with neuroscience labs and tool developers to support NWB. With the shift in focus from development to adoption then also came further refinement of the release processes and adoption of stricter software versioning guidelines based on semantic versioning to facilitate integration of NWB software with other software tools and adoption in lap data pipelines. While software releases in this phase were still often determined on a per-need-basis, the goal was to keep the APIs and standard as stable as possible.

One strategy to achieve this goal then was to separate the core data modeling capabilities from PyNWB into the separate HDMF library. Publishing HDMF as its own software product has been essential both to facilitate reuse of HDMF capabilities for other applications as well as to ensure stability of the PyNWB user API. As shown in *Appendix 7—figure 1*, PyNWB has undergone only 1 major release and 5 minor releases since the first release of PyNWB 1.0.0. At the same time, HDMF underwent a much larger number of releases. This illustrates the effectiveness of the approach of separating core infrastructure from user-APIs, as it allowed us to continue to advance core NWB technologies while limiting impact on end users. Similarly, extracting general schema (e.g., for dynamic data tables) into the separate hdmf-common-schema allowed to further make these common building blocks broadly accessible to science applications and to continue to develop them as part of the HDMF core software infrastructure.

MatNWB, through its strategy to auto-generate API classes directly from the NWB schema, is tied directly in a particular release to the most recent version of the NWB schema that the particular release supports. In April 2020, MatNWB, therefore, adopted a new, extended semantic versioning scheme for its software release consisting of 4 digits, with the first 3 digits indicating the major, minor, and patch release of the NWB schema and the last digit indicating the software patch release of MatNWB. As such, version 2.2.5.1 of MatNWB supports NWB schema 2.2.5 as the most recent version of the NWB format and includes 1 software patch release of MatNWB. MatNWB then also added all tagged versions of the NWB schema with each release to avoid the need for git checkouts during the use of the software and facilitate interaction with NWB files with varying schema versions.

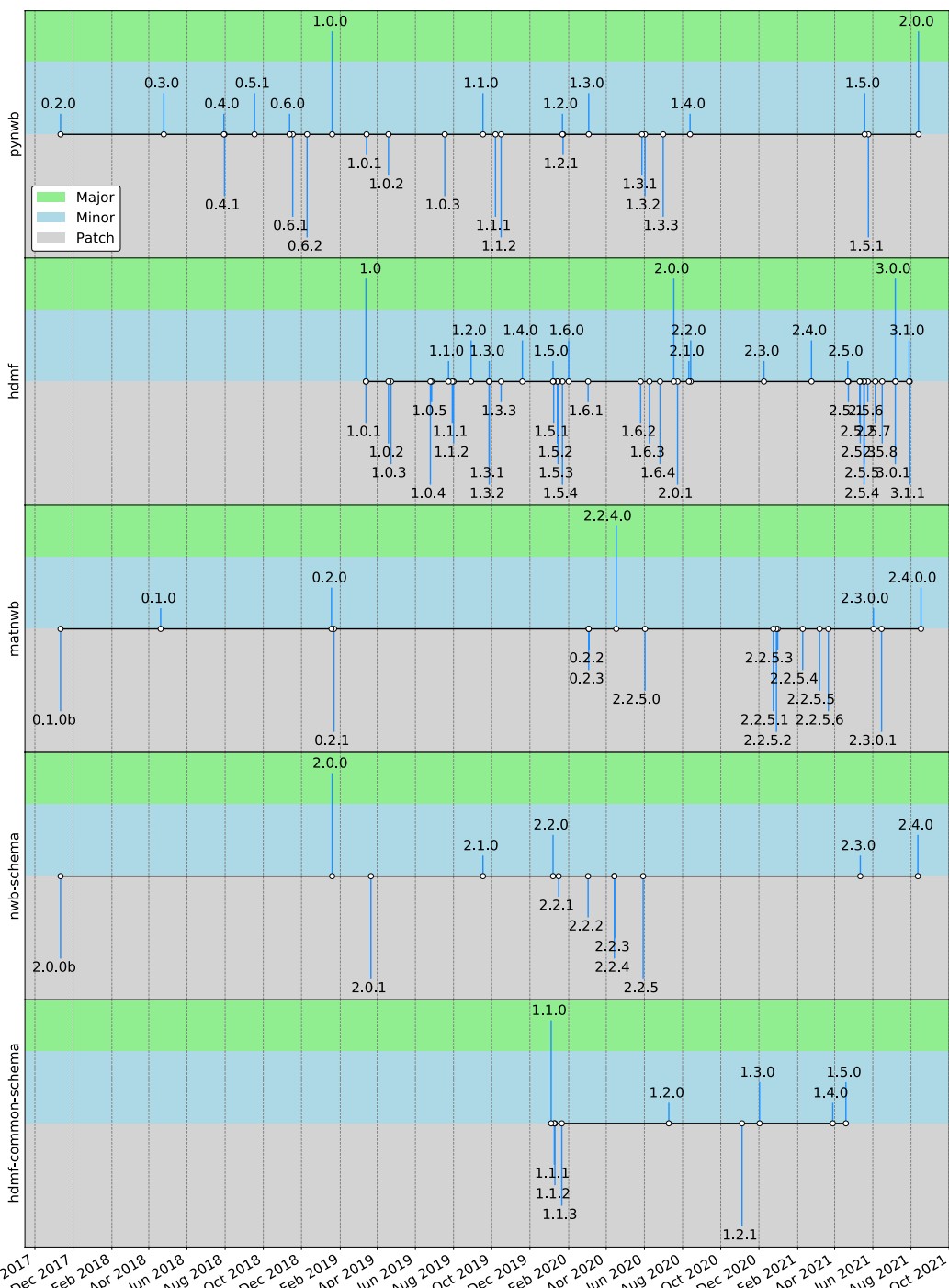

**Appendix 7—figure 1.** Software Release Process and History. Overview of the release history of the PyNWB, HDMF, and MatNWB APIs and the NWB and hdmf-common data standard schema.

With the start of the NIH U24 project in April 2021, NWB then entered its next main phase with a focus on more widespread adoption to advance standardization of neurophysiology data through dissemination and integrating of NWB. With this transition, then also comes the need for further refinement of software release processes. This also means that adoption and integration projects increasingly no longer involve the NWB team directly, but are being led independently by other project teams. To facilitate planning and interaction, this required further refinement of release processes to adopt more rigid release plans and schedules and to facilitate contribution of other projects to NWB with predictable release timelines.

In addition to the software, key components are also releases of the NWB schema. Here, a main goal has been stability to ensure that files remain accessible. This is also reflected in the release history of the NWB schema, which has undergone only four minor releases and no major releases since the first full release of the schema. These releases largely focused on addition and refinement of data schema, while the APIs support reading of data of all NWB 2 .x file versions.

## Appendix 8

### NWB Online and Social Media Resources

NWB online and social media resources provide additional resources for users and the broader community to engage with and learn about NWB. The nwb.org website serves as the central entry-point for users to NWB and provides high-level information about NWB and links to all relevant online resources and tools discussed in the Methods. Additional online resources include Slack (*NWB Slack, 2022*), Twitter (*NWB Twitter, 2022*), YouTube (*NWB YouTube, 2021*), and the NWB Mailing List (*NWB Mailing List, 2021*).

## Appendix 9

### NWB Training Resources

NWB provides a broad range of training resources for users and developers. Users who want to learn more about how to use NWB can view the NWB online video training course as part of the **[INCF Training Space]** (*INCF Training, 2022*). Detailed code tutorials are further available as part of the **[PyNWB Documentation]** (*PyNWB-b, 2021*) and **[MatNWB Documentation]** (*matnwb, 2021a*).

For Neurodata Extensions (NDX), detailed documentation of versioning guidelines, sharing guidelines and strategies, and the proposal review process are available online as part of the NDX Catalog (*NDXCatalog-a, 2022*). Step-by-step instructions for creating new NDX are provided as part of the [NWB Extensions Template] (*NDXtemplate, 2022*).

Additional resources for developers and data managers include the API documentation for [HDMF] (*HDMF-b, 2022*), [PyNWB] (*PyNWB-b, 2021*), and [MatNWB] (*matnwb, 2021a*) and documentation of the format schema as part of the [NWB Schema] (*NWB Schema-b, 2021*), [HDMF Common Schema] (*hdmf-common-schema-b, 2022*), [NWB Storage] (*NWB Storage, 2021*), and [Specification Language] (*NWB Specification Language, 2022*).

