## [Editor Report]

This manuscript provides an overview of an important project that proposes a common language to share neurophysiology data across diverse species and recording methods, Neurodata Without Borders (NWB). The NWB project includes tools for data management, analysis, visualization, and archiving, which are applicable throughout the context of the entire data lifecycle. This paper will help raise awareness of this endeavor and should be useful for many researchers across a broad range of fields who are interested in analyzing diverse neurophysiology datasets.

---

## [Decision Letter]

**Decision letter after peer review:**

[Editors’ note: the authors submitted for reconsideration following the decision after peer review. What follows is the decision letter after the first round of review.]

Thank you for submitting the paper "The Neurodata Without Borders ecosystem for neurophysiological data science" for consideration by *eLife*. Your article has been reviewed by 3 peer reviewers, and the evaluation has been overseen by a Senior Editor.

Comments to the Authors:

We are sorry to say that, after consultation with the reviewers, we have decided that this work will not be considered further for publication by *eLife*.

Reviewers agreed that Neurodata Without Borders (NWB) has merit and that the field will benefit greatly from this type of initiative. However, reviewers agreed that the paper in its present form reads more like a press release or an advertisement than a technical paper. Reviewers agreed that this language and style is inappropriate for an *eLife* paper. Reviewers felt that a more appropriate focus would have been to explain how NWB works and which advantages it provides over previous attempts at data standardization- to compare it more fairly with other similar attempts- rather than to advertise their success in an unbalanced manner. The paper lacks proof-of-concept of the advantages of this system and lacks benchmarks against existing or alternative systems/platforms. The technical innovations of this endeavor were unclear in this paper. It is unclear what makes NWB unique and novel and why scientists should adopt this format over other possible formats. Reviewers felt that the impact of this endeavor will be greatly limited if it is not broadly adopted, and this paper was not viewed as providing a compelling case as to why a diverse range of neuroscientists should adopt this platform.

*Reviewer #1:*

In general, providing new resources to enable the principles of FAIR data in the neuroscience community is an important and worthwhile effort. It has been addressed by many national funding and research institutions through issuing clear guidelines, summarized in Data Management Plans. Many labs have implemented this and their publications give clear instructions on how to access the source data. It seems likely that data searching and sharing will always heavily rely on reading scientific papers and interacting personally with the authors to discuss the datasets. Still, to have a common archive for a larger community can further simplify exchange and sharing of data, and also guide the design of novel experiments.

However, I currently do not think that this manuscript meets the standards for Tools and Resources article in a scientific journal as *eLife*. The article is a peculiar mix of meta-language that reads more as an advertisement rather than as a way to provide "exploratory or proof-of-concept" experiments on a new methodology. Also, there are statements about past efforts in the field that, in my view, do not adequately represent the situation and thus cannot be accepted as a "thorough benchmark against existing technology". Finally, the writing contains Jargon from computer language and over-synthesized figures that make me think that the manuscript should be edited by a neuroscientist for better access to the target public. There are additional major gaps, such as clear guidelines for the researchers and explicit statements of the added values of adding data to NWB, and a plan on whether/how data will be safeguarded and curated in the mid- and long-term future.

1) On p.6, it is said that data become useless when the individual who generated that data leaves the lab. I do not think that this adequately represents the current situation given that many labs tightly follow data management plans in which this very problem is addressed.

2) I do not understand the choice of experiments used in Figure 1 that seems rather arbitrary. As a Tools and Resources article, I am not sure whether such loose assembly figures that seem mostly there to provide a visual illustration are really necessary.

3) The abstract should state that all software presented here is open-source.

4) The work cites previous efforts to organize large-scale data in the context of the US brain initiative or the Human Brain Project. Then it is asked why these were "not successful". I do not understand this question because it implies that being successful means only being used by many people. This is not an appropriate representation of the success of these efforts.

5) As an experimental neuroscientist, I do not understand the description of current data as "standard, monolithic" (p.8). I find this an unnecessary pejorative characterization.

6) At the same place, I find the wording of a "conceptual departure from a traditional notion of a … data standard" one of the many, many examples of a metadata language that relies on unsubstantiated wording without providing the reader with useful information.

7) In a Tools and Resources article, is it really necessary to speak about prizes and recognitions of significance? The entire text around Figure 2 would be more suited for a flyer or home page advertisement of a technology rather than for an article in a scientific journal aiming to provide tools and resources to the readers.

8) What justifies the word "ecosystem"? What is the factual evidence for it?

9) Every new subchapter in the Results part re-initiates on basic considerations circling around challenges of data organization, heterogeneity of data, multiple streams of data, generating data at an unprecedented scale…. I think these considerations should be part of the introduction and it would be sufficient to declare them once. Then, the introduction could also be freed of even more basic considerations on the "immense diversity of life on earth".

10) As indicated in the guidelines for Tools and Resources, the manuscript should provide exploratory or proof-of-concept examples/experiments demonstrating that NWB has provided real advances and new biological insights because of the facilitated access to data.

11) What is the decision tree the researcher should go through before deciding to upload his data to NWB? What are the legal constraints? Can NWB replace local repositories and who takes the responsibility for stored data? How is appropriate referencing done once data uploaded by one researcher are downloaded by another one?

12) There should be quantification of data uploading and sharing between current users and an evaluation on how this has affected their research.

13) There should be a discussion of the current limits of NWB, limits from the simple indication of data size up to the fact that often nowadays neurophysiology is combined with multiple parallel analysis, ranging from genetics to varied behavioral manipulations that include animal handling, housing, husbandry, circadian entrainment, etc.

*Reviewer #2:*

This work details the ways in which the developed Neurodata Without Borders (NWB) ecosystem tackles the challenges of Findability, Accessibility, Interoperability, and Reusability (FAIR) for neurophysiological data. From a technical standpoint, this work clearly describes how NWB can serve as a flexible but sustainable data standard for neurophysiological data. This work discusses the software stack of NWB, the individual components, and how users and developers can interact with the ecosystem, and provides a high-level overview of how data can be interoperable between common programming languages used in neuroscience. The work also describes the ability of the NWB standard to accommodate multimodal neural and behavioral data, with infrastructure in place to allow for the integration of novel experiment paradigms and data formats. The authors describe internal- and community-based review processes to potentially integrate these novel formats into the core of the NWB standard. Furthermore, this work discusses existing integration with certain data acquisition systems and current efforts to expand the list of compatible acquisition systems and describes tools and efforts to convert data from legacy systems. Finally, this work describes a developed data repository, DANDI, that uses the NWB standard, and how it overcomes existing issues with current repositories in the context of findability and reusability. From a sociological standpoint, this work shows evidence of its growing user base and adoption with references to further efforts to increase adoption through events, workshops, resources, etc. Overall, this is not a traditional scientific paper, and is rather a description of the NWB platform that is meant to raise awareness among neuroscience researchers.

Details and Comments:

To expand on the components, for the purposes of data modeling and standardization, the authors developed the NWB format schema and specification language, which are both based on the existing Hierarchical Data Modeling Framework (HDMF). For the purpose of data storage, they have developed software that translates NWB-formatted data primitives to common backend storage formats such as HDF5. For the translation and use of data by end-users, they have developed application programming interfaces (APIs) for both matlab and python. These APIs allow end-users to load, visualize, analyze, and save NWB-formatted data. These two APIs are also interoperable, meaning that files created in one can be read by the other, thereby allowing increased collaboration between organizations that use different programming languages. In order to allow the neuroscience community to extend the functionality of NWB, the authors have released a set of templates and tools that allow for the development of Neurodata Extensions (NDX) by end-users. These extensions allow users to add support for specific types of data that may not be supported by the NWB core. Lastly, the authors have developed the Distributed Archives for Neurophysiology Data Integration (DANDI), a web-based data archive for NWB datasets. This archive is currently in early-access, and contains 18 TB of data across 71 datasets.

The authors have also developed NWB's organizational structure, as well as community outreach in order to increase the adoption of NWB by the neuroscience community. The authors have developed an organizational structure for the purposes of data governance and community engagement, which will promote the longevity and sustainability of their platform. They have additionally developed a formal review process whereby community members can suggest changes to the NWB core, which will ideally allow NWB to evolve with the needs of the community. The authors have shown that their outreach efforts have so far been successful by listing scientists and institutions that have adopted the NWB format and have contributed to the DANDI data archive.

In summary, the authors have described the goals and functionality of the NWB ecosystem, and have shown that its adoption by the neuroscience committee is promising. The primary purpose of this work seems to be to raise awareness of the NWB platform in the neuroscience community, and the authors have demonstrated its viability as a unifying framework that has the potential to improve collaboration between neuroscientists. The success of NWB as a unifying platform will ultimately depend on whether the broader community will further embrace and adopt it.

I believe the paper sufficiently describes the goals and approaches of NWB. My only main comment for the authors is to clarify the technical/technological innovations of this work better, compared to previous attempts at data standardization within the neuroscience community, if any. The authors do not seem to mention any other data standards by name other than their previous NWB version. Are there any technological innovations here compared to previous attempts? Or is it that there have been no previous coordinated attempts at this scale for writing and organizing the software such that it can facilitate future additions, and mechanisms that allow this work to evolve with what's new in the field? Is the innovation mainly in the software to facilitate these aspects and community engagement?

*Reviewer #3:*

The manuscript starts by outlining the motivation, development and dissemination of NWB. These sections are interesting, but they extend too long and include excessive digressions into rather philosophical or historical aspects that dilute the main message of the paper. I believe some of this space would be better used to describe in more detail other technical aspects of NWB. The following sections describe with some level of detail the different components of the NWB 'ecosystem'. While it is understandable that such a large project cannot be fully described in a paper format, I would be useful to expand the description (perhaps in supplemental material) of some concrete examples of its application (as the one described in Figure 4). The last section described a repository of NWB dataset, DANDI, aimed at collaborative research. This is an important complement to the NWB pipeline; however, it is not completely clear how does it differ from other existing and successful repositories for neurophysiology (e.g. CRCNS). In this regard, vague statements (e.g. 'few data archives today support a collaborative research model') should be substituted by specific arguments.

The manuscript is well articulated and clearly explains the motivation and implementation of NWB in an accessible manner for the general reader. However, it has, in my opinion, a significant flaw. I believe that the goal of this paper should be not only to showcase the impressive progress of the NWB initiative, but also to convince other researchers to adopt it. While the former was largely achieved, a better effort could be done towards the later. The cost of adopting a new data standard is quite large for an individual laboratory. The NWB community has done an excellent effort to ease this process, through workshops, tutorials, etc. On the other hand, they still have in the first place to convince individual researchers that adopting NWB is worth the effort. In this regard, the description of NWB adoption seems overstated in the paper. While it is impressive the reach shown in hackathons and workshops, the labs in Table 1 have perhaps contributed with some test datasets to NWB or participated somehow in the initiative, but are not using NWB as their internal data standard (at least not many of them). In my view, a way of achieving this, would be to offer and array of tools from which labs adopting NWB can immediately benefit. There are many examples in Neuroscience and beyond of data analysis toolboxes that have been tremendously successful and wide-spread and a data format was adopted as a consequence of this. I believe that, if NWB would put a stronger emphasis in the development of tools around their data format, that will offer a better incentive for many potential adopters. In the manuscript, the description of such tools or plans for their future development is nearly absent, with the exception of tools for converting data formats or visualizing NWB files. The tools described to manipulate and visualize NWB are an important development, but in themselves do not offer much incentive for would-be adopters. This is a long process and perhaps beyond the scope of the current manuscript. What should be done in the present manuscript, is to more clearly outline the short-term advantages for individual researchers, especially those generating data, of adopting NWB. So far the long-term advantages for the broad community or the immediate advantages for scientist looking for open datasets to analyze are clear. But if NWB is really to succeed it needs to also convince the individual groups and researchers that already work with their own data formats and use other platforms to share their data, that adopting NWB has unique and concrete advantages for their daily research activities.

---

## [Author Response]

[Editors’ note: The authors appealed the original decision. What follows is the authors’ response to the first round of review.]

Comments to the Authors:We are sorry to say that, after consultation with the reviewers, we have decided that this work will not be considered further for publication by eLife.Reviewers agreed that Neurodata Without Borders (NWB) has merit and that the field will benefit greatly from this type of initiative. However, reviewers agreed that the paper in its present form reads more like a press release or an advertisement than a technical paper. Reviewers agreed that this language and style is inappropriate for an eLife paper. Reviewers felt that a more appropriate focus would have been to explain how NWB works and which advantages it provides over previous attempts at data standardization- to compare it more fairly with other similar attempts- rather than to advertise their success in an unbalanced manner. The paper lacks proof-of-concept of the advantages of this system and lacks benchmarks against existing or alternative systems/platforms. The technical innovations of this endeavor were unclear in this paper. It is unclear what makes NWB unique and novel and why scientists should adopt this format over other possible formats. Reviewers felt that the impact of this endeavor will be greatly limited if it is not broadly adopted, and this paper was not viewed as providing a compelling case as to why a diverse range of neuroscientists should adopt this platform.Reviewer #1:In general, providing new resources to enable the principles of FAIR data in the neuroscience community is an important and worthwhile effort. It has been addressed by many national funding and research institutions through issuing clear guidelines, summarized in Data Management Plans. Many labs have implemented this and their publications give clear instructions on how to access the source data. It seems likely that data searching and sharing will always heavily rely on reading scientific papers and interacting personally with the authors to discuss the datasets. Still, to have a common archive for a larger community can further simplify exchange and sharing of data, and also guide the design of novel experiments.However, I currently do not think that this manuscript meets the standards for Tools and Resources article in a scientific journal as eLife. The article is a peculiar mix of meta-language that reads more as an advertisement rather than as a way to provide "exploratory or proof-of-concept" experiments on a new methodology. Also, there are statements about past efforts in the field that, in my view, do not adequately represent the situation and thus cannot be accepted as a "thorough benchmark against existing technology". Finally, the writing contains Jargon from computer language and over-synthesized figures that make me think that the manuscript should be edited by a neuroscientist for better access to the target public. There are additional major gaps, such as clear guidelines for the researchers and explicit statements of the added values of adding data to NWB, and a plan on whether/how data will be safeguarded and curated in the mid- and long-term future.1) On p.6, it is said that data become useless when the individual who generated that data leaves the lab. I do not think that this adequately represents the current situation given that many labs tightly follow data management plans in which this very problem is addressed.

Per the reviewer's suggestion, we have updated the text to instead state: “As such, even within the same laboratory, representations of data and metadata often vary significantly between experiments, making sharing and reuse of data a significant challenge.”

2) I do not understand the choice of experiments used in Figure 1 that seems rather arbitrary. As a Tools and Resources article, I am not sure whether such loose assembly figures that seem mostly there to provide a visual illustration are really necessary.

These choices were selected to exemplify the diversity of biological and experimental conditions in neurophysiology, and largely reflect the authors on the paper. As our Introduction describes, this diversity is at the heart of the challenge of creating a unified data language for neurophysiology data. As *eLife* is a general biology venue, read by not just neuroscientists, we feel it is important to provide readers an overview of the biological and experimental NWB contends with.

More broadly, the same conceptual and technical challenges to standardization and FAIRness present in neurophysiology data is present in other biological fields. In our many interactions with biologists in other fields (e.g., synthetic biology, genomics, medical science, etc.,) we experience a persistent misconception that the diversity within those fields prevents meaningful standardization. Thus, as we are already utilizing the data science concepts and software tools developed under NWB in other biological fields, it is important to communicate that data science solutions in one biological field can be used (with modifications) to address similar issues in other disciplines of biology.

3) The abstract should state that all software presented here is open-source.

Per the reviewers suggestion, we have added “open source” to the Abstract: “Our open-source software (Neurodata Without Borders, NWB) defines and modularizes the interdependent, yet separable, components of a data language.”

4) The work cites previous efforts to organize large-scale data in the context of the US brain initiative or the Human Brain Project. Then it is asked why these were "not successful". I do not understand this question because it implies that being successful means only being used by many people. This is not an appropriate representation of the success of these efforts.

We thank the reviewer for bringing this issue to our attention. We broadly agree that there are multiple definitions of success, and that adaptation, while an important one (see comments by Reviewers 2 and 3), is not the only one. We have therefore removed this part of the Introduction as part of our broader reorganization. The comparison to other technologies in the context of FAIR data principles is now provided in the new Results section NWB and DANDI build the foundation for a FAIR neurophysiology data ecosystem, and corresponding Table 1, and in Supplementary Material 6. Diverse Community of Data Producers Adopting NWB.

5) As an experimental neuroscientist, I do not understand the description of current data as "standard, monolithic" (p.8). I find this an unnecessary pejorative characterization.6) At the same place, I find the wording of a "conceptual departure from a traditional notion of a … data standard" one of the many, many examples of a metadata language that relies on unsubstantiated wording without providing the reader with useful information.

We apologize for the lack of clarity in our writing on this point. We wish to clarify that it was not ‘current data’ that we were referring to as “monolithic”, but the conceptualization of ‘data standards’ in the community as such, None-the-less, per the reviewer's suggestion, we have changed the term “monolithic” to “rigid and static” and expanded the paragraph to clarify the typical use-case for this approach and why we need a different strategy here. The updated text, found on Pg. 5 of the Introduction, is:

“Traditionally, data standards are often understood as rigid and static data models and formats. Such standards are particularly useful to enable the exchange of specific data types (e.g., image data), but are insufficient to address the diversity of data types generated by constantly evolving experiments. Together, these challenges and requirements necessitate a conceptual departure from the traditional notion of a rigid and static data standard. That is, we need a “language” where fundamental structures can be reused and combined in new ways to express novel concepts and experiments. A data language for neurophysiology will enable precise communication about neural data that can co-evolve with the needs of the neuroscience community.”

7) In a Tools and Resources article, is it really necessary to speak about prizes and recognitions of significance? The entire text around Figure 2 would be more suited for a flyer or home page advertisement of a technology rather than for an article in a scientific journal aiming to provide tools and resources to the readers.

We take the reviewers' request to be a removal of the awards to NWB in the Figure (now Figure 7), as well as in the corresponding text. We have done so in the new Figure 7 and associated text.

However, we disagree with the characterization of the Figure (now Figure 7) as being “more suited for a flyer or home page advertisement of a technology rather than for an article in a scientific journal aiming to provide tools and resources to the readers.” As is well known by people involved with data standardization efforts, a major challenge is the social-engineering aspect of engaging with the community to ensure software tools are aligned with user needs, and to get the community to use the software. Indeed, as pointed out by Reviewer 3, adoption of NWB is a major metric of success. Figure 7 and the accompanying text directly address the issues of community engagement and governance, as well as how software development interacts with user engagement activities, and the growing international interest in NWB. As adoption is a key metric of success for a data standard, we believe it is important to highlight this.

8) What justifies the word "ecosystem"? What is the factual evidence for it?

We thank the reviewer for pointing out the lack of clarity in this concept. We added the following paragraph to the introduction to more clearly define the term ecosystem as used in the paper. Specifically, from the Introduction, pg.6:

“Scientific data must be thought of in the context of the entire data lifecycle, which spans planning, acquisition, processing, and analysis to publication and reuse. In this context, a “data ecosystem” is a shared market for scientific data, software, and services that are able to work together. Such an ecosystem for neurophysiology would empower users to integrate software components and products from across the ecosystem to address complex scientific challenges. Foundational to realizing a data ecosystem is a common ‘language’ that enables seamless exchange of data and information between software components and users. Here, the principles of Findable, Accessible, Interoperable, and Reusable (i.e., FAIR)data management and stewardship are widely accepted as essential to ensure that data can flow reliably between the components of a data ecosystem.”

The factual evidence for an “NWB ecosystem” is described in detail in the various sections of the Results portion of the manuscript, and in particular two sections.

“NWB is integrated with state-of-the-art analysis tools throughout the data life cycle” and the associated Figure 6 (Pg. 21-23 of the Results). This section/Figure 6, we show how NWB enables diverse scientific data (e.g, optical physiology, electrophysiology, etc.,), software (e.g., SpikeInterface, CalmAn, suite2p), and services (e.g., DataJoint, DABI, DANDI) to work together.

“NWB and DANDI build the foundation for a FAIR neurophysiology data ecosystem.” And the corresponding Table 1. This section/Table describes how, together, NWB and DANDI provide the neurophysiology community with the tools required to meaningfully share data in a FAIR ecosystem. This is further expanded upon in Supplementary Material 5.

9) Every new subchapter in the Results part re-initiates on basic considerations circling around challenges of data organization, heterogeneity of data, multiple streams of data, generating data at an unprecedented scale…. I think these considerations should be part of the introduction and it would be sufficient to declare them once. Then, the introduction could also be freed of even more basic considerations on the “immense diversity of life on earth”.

We thank the reviewer for pointing out this redundancy. Per the reviewer’s suggestion, we have removed this redundancy from the Results sections and stated it clearly in the Introduction (pg.1-2), illustrated by examples in Figure 1. We have also streamlined the Introduction and reorganized the Results section.

10) As indicated in the guidelines for Tools and Resources, the manuscript should provide exploratory or proof-of-concept examples/experiments demonstrating that NWB has provided real advances and new biological insights because of the facilitated access to data.

We believe that our manuscript does provide examples demonstrating that NWB provides real advances. For a technology such as NWB, the goals are 2-fold: to enable unified storage of diverse data, derived products, and associated metadata; make neurophysiology data more FAIR. For the first point, in Figure 2, we provide an example of storing raw electrophysiology and optical physiology data, as well as the derived products, and expanded upon the associated text. We have additionally included a new example of NWB storage and visualization of intracellular electrophysiology data from DANDI. The basic code and visualizations are provided in Supplementary Material 1. Intracellular Electrophysiology Example using NWB and DANDI. For the second point, we have additionally included an entire new section in the Results (pg.23-25) NWB and DANDI build the foundation for a FAIR neurophysiology data ecosystem, with new Table 1 which summarizes a comparison of NWB+DANDi relative to other technologies for making neurophysiology data FAIR. This analysis is expanded upon in further detail in Supplementary Material 5. Assessment of FAIRness of NWB + DANDI.

11) What is the decision tree the researcher should go through before deciding to upload his data to NWB? What are the legal constraints? Can NWB replace local repositories and who takes the responsibility for stored data? How is appropriate referencing done once data uploaded by one researcher are downloaded by another one?

We take the reviewers request to be centered around issues of data sharing repositories. This is directly addressed by the section NWB is foundational for the DANDI data repository to enable collaborative data sharing, on Pg. 17. To briefly answer the reviewers specific queries: We note that the NIH is now requiring NIH funded data to be stored and shared in their accepted formats and repositories, such as NWB and DANDI. Data sharing of NWB formatted data can be provided by several repositories, all of which have their own legal constraints (see, e.g., https://dandiarchive.org). NWB does not replace repositories, but makes the data stored in those open repositories more FAIR. DANDI links NWB Dandisets to eventual publications when generated via DOIs.

12) There should be quantification of data uploading and sharing between current users and an evaluation on how this has affected their research.

We take the reviewers comments to be a request for quantification of data uploading to a shared repository that uses NWB. This is directly provided in two parts of our manuscript. In NWB is foundational for the DANDI data repository to enable collaborative data sharing, on Pg. 17, and the corresponding Figure 5, we show that there are 138 NWB formatted data sets, 311 users, and 157 TBs of data. Additionally, in Supplementary Material 6. Diverse Community of Data Producers Adopting NWB, we provide a detailed list of more than 50 neurophysiology labs that have either submitted NWB formatted datasets to DANDI, or for which we have directly worked with to create conversion pipelines for their data.

13) There should be a discussion of the current limits of NWB, limits from the simple indication of data size up to the fact that often nowadays neurophysiology is combined with multiple parallel analysis, ranging from genetics to varied behavioral manipulations that include animal handling, housing, husbandry, circadian entrainment, etc.

We agree with the reviewer that discussing the limits of NWB is important, and we have now done so in the Discussion section “The Future of NWB” on pg. 31.

Reviewer #2:[…]I believe the paper sufficiently describes the goals and approaches of NWB. My only main comment for the authors is to clarify the technical/technological innovations of this work better, compared to previous attempts at data standardization within the neuroscience community, if any. The authors do not seem to mention any other data standards by name other than their previous NWB version. Are there any technological innovations here compared to previous attempts? Or is it that there have been no previous coordinated attempts at this scale for writing and organizing the software such that it can facilitate future additions, and mechanisms that allow this work to evolve with what’s new in the field? Is the innovation mainly in the software to facilitate these aspects and community engagement?

We thank the reviewer for suggesting this analysis. We wholeheartedly agree, and we have included an entire new section in the Results (pg.23-25) “NWB and DANDI build the foundation for a FAIR neurophysiology data ecosystem”, with new Table 1 which summarizes a comparison of NWB+DANDi relative to other technologies for making neurophysiology data FAIR. This analysis is expanded upon in further detail in Supplementary Material 5. Assessment of FAIRness of NWB + DANDI. We have also expanded the Results section “The NWB software architecture modularizes and integrates all components of a data language.” And provided examples of code in the Supplement. We believe the inclusion of this new analysis and material directly addresses the reviewers’ concerns and greatly enhances our manuscript.

Reviewer #3:The manuscript starts by outlining the motivation, development and dissemination of NWB. These sections are interesting, but they extend too long and include excessive digressions into rather philosophical or historical aspects that dilute the main message of the paper. I believe some of this space would be better used to describe in more detail other technical aspects of NWB. The following sections describe with some level of detail the different components of the NWB ‘ecosystem’.While it is understandable that such a large project cannot be fully described in a paper format, I would be useful to expand the description (perhaps in supplemental material) of some concrete examples of its application (as the one described in Figure 4).

We thank the reviewer for suggesting the importance of providing expanded descriptions of specific applications. We agree that it is important to emphasize that NWB has already been successfully applied to diverse neurophysiology data sets. As part of our effort to enhance this aspect of our work, we have moved the example of storing raw electrophysiology and optical physiology data, as well as the derived products, up to Figure 2 (previously Figure 4) and expanded upon the associated text. We have additionally included a new example of NWB storage and visualization of intracellular electrophysiology data from DANDI. The basic code and visualizations are provided in Supplementary Material 1. Intracellular Electrophysiology Example using NWB and DANDI.

The last section described a repository of NWB dataset, DANDI, aimed at collaborative research. This is an important complement to the NWB pipeline; however, it is not completely clear how does it differ from other existing and successful repositories for neurophysiology (e.g. CRCNS). In this regard, vague statements (e.g. ‘few data archives today support a collaborative research model’) should be substituted by specific arguments.

We thank the reviewer for pointing out the lack of clarity surrounding the DANDI repository in how it differs from existing repositories. We have addressed this issue in several ways. We have expanded the text associated with that section to make the contrast with existing repositories more direct. Specifically, from pg. 17-18 of the Results section NWB is foundational for the DANDI data repository to enable collaborative data sharing:

“Making neurophysiology data accessible supports published findings and allows secondary reuse. To date, many neurophysiology datasets have been deposited into a diverse set of repositories (e.g., CRCNS, Figshare, Open Science Framework, Gin). However, no single data archive provides the neuroscientific community the capacity and the domain specificity to store and access large neurophysiology datasets. Most current repositories have specific limits on data sizes and are often generic, and therefore lack the ability to search using domain specific metadata. Further, for most neuroscientists, these archives often serve as endpoints associated with publishing, while research is typically an ongoing and collaborative process. Few data archives support a collaborative research model that allows data submission prospectively, analysis of data directly in the archive, and opening the conversation to a broader community. Enabling reanalysis of published data was a key challenge identified by the BRAIN Initiative. Together, these issues impede access and reuse of data, ultimately decreasing the return on investment into data collection by both the experimentalist and the funding agencies.”

We have additionally included an entire new section in the Results (pg.23-25) “NWB and DANDI build the foundation for a FAIR neurophysiology data ecosystem”, with new Table 1 which summarizes a comparison of NWB+DANDi relative to other technologies for making neurophysiology data FAIR. This analysis is expanded upon in further detail in Supplementary Material 5. Assessment of FAIRness of NWB + DANDI.

The manuscript is well articulated and clearly explains the motivation and implementation of NWB in an accessible manner for the general reader. However, it has, in my opinion, a significant flaw. I believe that the goal of this paper should be not only to showcase the impressive progress of the NWB initiative, but also to convince other researchers to adopt it. While the former was largely achieved, a better effort could be done towards the later. The cost of adopting a new data standard is quite large for an individual laboratory. The NWB community has done an excellent effort to ease this process, through workshops, tutorials, etc. On the other hand, they still have in the first place to convince individual researchers that adopting NWB is worth the effort. In this regard, the description of NWB adoption seems overstated in the paper. While it is impressive the reach shown in hackathons and workshops, the labs in Table 1 have perhaps contributed with some test datasets to NWB or participated somehow in the initiative, but are not using NWB as their internal data standard (at least not many of them). In my view, a way of achieving this, would be to offer and array of tools from which labs adopting NWB can immediately benefit. There are many examples in Neuroscience and beyond of data analysis toolboxes that have been tremendously successful and wide-spread and a data format was adopted as a consequence of this. I believe that, if NWB would put a stronger emphasis in the development of tools around their data format, that will offer a better incentive for many potential adopters. In the manuscript, the description of such tools or plans for their future development is nearly absent, with the exception of tools for converting data formats or visualizing NWB files. The tools described to manipulate and visualize NWB are an important development, but in themselves do not offer much incentive for would-be adopters. This is a long process and perhaps beyond the scope of the current manuscript. What should be done in the present manuscript, is to more clearly outline the short-term advantages for individual researchers, especially those generating data, of adopting NWB. So far the long-term advantages for the broad community or the immediate advantages for scientist looking for open datasets to analyze are clear. But if NWB is really to succeed it needs to also concince the individual groups and researchers that already work with their own data formats and use other platforms to share their data, that adopting NWB has unique and concrete advantages for their daily research activities.

We agree with the reviewer that part of our goal is to convince researchers to adopt NWB. We do this in several ways throughout the manuscript by showing that (1) diverse neurophysiology data and derived products can be stored in NWB; (2) show how NWB is seamlessly integrated into diverse analysis software packages; (3) that NWB is intimately connected with the neuroscience community and provides lots of diverse opportunities for training and outreach; (4) we have expanded discussion of on-going work.

In more detail:

1) We provide examples of storage of diverse neurophysiology data and derived products in NWB (Results section NWB enables unified description and storage of multimodal data and derived products, and Supplementary Material 1. Intracellular Electrophysiology Example using NWB and DANDI). These results provide real examples that potential future users can see that their types of neurophysiology data are already storable in NWB.

2) We show how NWB is seamlessly integrated into diverse analysis software packages, such as SpikeInterface, CalmAN, suite2p, ipfx, etc. as well as other data acquisition software (e.g., OpenEphys) and data management systems (e.g., DataJoint). This is done in the Results section NWB is integrated with state-of-the-art analysis tools throughout the data life cycle. This shows that the data analysis community is building around NWB and that common existing tools support NWB.

3) We show that NWB is intimately connected with the neuroscience community and provides lots of diverse opportunities for training and outreach. This is done in the Results section Coordinated Community Engagement, Governance, and Development of NWB.

4) Finally, per the reviewers suggestion, we have expanded discussion of on-going work. Specifically, from pg. 32 of the Discussion section The Future of NWB:

“As adoption of NWB continues to grow, new needs and opportunities for further harmonization of metadata arise. A key ongoing focus area is on development and integration of ontologies with NWB to enhance specificity, accuracy, and interpretability of data fields. For example, there are NWB working groups on genotype and spatial coordinate representation, as well as the INCF Electrophysiology Stimulation Ontology working group. Another key area is extending NWB to new areas, such as the ongoing working groups on integration of behavioral task descriptions with NWB (e.g., based on BEADL_49_) and enhanced integration of simulations with NWB. We strongly advocate for funding support of all aspects of the data-software life cycle (development, maintenance, integration, and distribution) to ensure the neuroscience community fully reaps the benefits of investment into neurophysiology tools and data acquisition.”